# Structural basis of ligand binding modes of human EAAT2

Zhenglai Zhang [1,2,7], Huiwen Chen[1,2,7], Ze Geng[3,4,7], Zhuoya Yu[2,5,6], Hang Li[2,6], Yanli Dong [2], Hongwei Zhang[2,6], Zhuo Huang [3,4✉], Juquan Jiang [1✉] & Yan Zhao [2,5,6✉]

In the central nervous system (CNS), excitatory amino acid transporters (EAATs) mediate the uptake of excitatory neurotransmitter glutamate and maintain its low concentrations in the synaptic cleft for avoiding neuronal cytotoxicity. Dysfunction of EAATs can lead to many psychiatric diseases. Here we report cryo-EM structures of human EAAT2 in an inward-facing conformation, in the presence of substrate glutamate or selective inhibitor WAY-213613. The glutamate is coordinated by extensive hydrogen bonds and further stabilized by HP2. The inhibitor WAY-213613 occupies a similar binding pocket to that of the substrate glutamate. Upon association with the WAY-213613, the HP2 undergoes a substantial conformational change, and in turn stabilizes the inhibitor binding by forming hydrophobic interactions. Electrophysiological experiments elucidate that the unique S441 plays pivotal roles in the binding of hEAAT2 with glutamate or WAY-213613, and the I464-L467-V468 cluster acts as a key structural determinant for the selective inhibition of this transporter by WAY-213613.

[1] Department of Microbiology and Biotechnology, College of Life Sciences, Northeast Agricultural University, No. 600 Changjiang Road, Xiangfang District, Harbin 150030, China. [2] National Laboratory of Biomacromolecules, CAS Center for Excellence in Biomacromolecules, Institute of Biophysics, Chinese Academy of Sciences, Beijing 100101, China. [3] State Key Laboratory of Natural and Biomimetic Drugs, Department of Molecular and Cellular Pharmacology, School of Pharmaceutical Sciences, Peking University Health Science Center, Beijing 100191, China. [4] IDG/McGovern Institute for Brain Research, Peking University, Beijing 100871, China. [5] State Key Laboratory of Brain and Cognitive Science, Institute of Biophysics, Chinese Academy of Sciences, 15 Datun Road, Beijing 100101, China. [6] College of Life Sciences, University of Chinese Academy of Sciences, Beijing 100049, China. [7] These authors contributed equally: Zhenglai Zhang, Huiwen Chen, Ze Geng. ✉email: huangz@hsc.pku.edu.cn; jjqdainty@163.com; zhaoy@ibp.ac.cn

Glutamate is the predominant excitatory neurotransmitter, which serves a key role in the development of the mammalian central nervous system, and participates in normal brain function, such as incorporating learning, cognition, and memory[1]. Also, excessive glutamate can lead to excitotoxicity, which may kill neuronal cells through excessive stimulation of glutamate receptors[2]. Thus, it is crucial to maintain low concentrations of glutamate in extracellular fluids. EAATs are known as excitatory amino acid transporters including five subtypes (EAAT1–EAAT5), which are responsible for the removal of glutamate from the synaptic cleft by rapidly binding and transporting glutamates back up into the pre-synapse or astrocytes, which contributes to the termination of synaptic activity and to the clearance of potentially cytotoxic extracellular glutamate[3]. Among the five EAATs, human EAAT2 (hEAAT2) is predominantly expressed in astrocytes and was reported to be responsible for 90–95% of glutamate uptake in the forebrain by an elevator mechanism[3,4]. Therefore, hEAAT2 can maintain extracellular glutamate at low levels in the synaptic cleft, which is one of the most important EAATs to protect neurons[5,6]. Deficiency of hEAAT2 causes progressive neuronal death, and psychiatric or neurological diseases, including major depressive disorder, epilepsy, Alzheimer's disease, stroke, Parkinson's disease, and amyotrophic lateral sclerosis (ALS), and thus hEAAT2 represents a potential therapeutic target[6–8].

EAATs are structurally similar to the archaeal homolog GltPh, all of which are homotrimers composed of three identical protomers[9,10]. The individual protomer has eight transmembrane helices (TM 1–8) and two hairpin loops (HP) organized into two domains: the scaffold domain comprises TM1, 2, 4, and 5, which is responsible for stabilizing the transport domain; the transport domain includes TM 3, 6, 7, and 8, HP1 and HP2. Previous studies have shown that the individual subunit in EAATs can transport the substrate independently[11–13], and the substrate-binding site is constituted by the central unwound region of TM7 (NMDGT motif), TM8 and tips of HP1 and HP2[3,9].

The structures of archaeal homologs (GltPh[4,9,10,14,15], GltTk[16–18]) and four human SLC1 transporters (hEAAT1[19], hEAAT3[20], hASCT1[21], and hASCT2[22–25]) have been solved, and they share high sequence conservation in the ligand-binding pocket including TM7, TM8, HP1, and HP2. However, it is still desirable to determine the structure of hEAAT2, as it shares low sequence identities with these homologs (34% sequence identity with GltPh and GltTk, 50% sequence identity with hEAAT1 and hEAAT3, 42% and 39% sequence identity with hASCT1 and hASCT2, respectively). WAY-213613 is a potent and highly selective inhibitor for hEAAT2 (IC$_{50}$ is 85 nM), which has 59-fold and 44-fold affinity over those of hEAAT1 and hEAAT3 (IC$_{50}$ is 5 and 3.8 μM, respectively)[26]. However, the mechanism that WAY-213613 highly selectively inhibits glutamate transport by hEAAT2 remains elusive.

Here, we report two cryo-EM structures of trimeric hEAAT2 in complex with glutamate and WAY-213613, respectively. Both of structures are determined at 3.4 Å resolution and stabilized at the inward-facing conformational state. These structures elucidate the structural basis of how the substrate is recognized by hEAAT2 and how the WAY-213613 selectively inhibits the transporter. The electrophysiological experiments were carried out and confirmed our speculations.

## Results

### Functional characterization and architecture of the hEAAT2.
In addition to mediating excitatory amino acid transport, hEAAT2 also acts as an anion-selective channel[27–29]. The hEAAT2-associated current includes three components: glutamate-induced anion current, Na$^+$-dependent anion leak current, and glutamate transport current[30,31]. The competitive inhibitors can block all three components of the current[32]. To gain more insights into glutamate and inhibitor binding sites, we carried out electrophysiological experiments using the whole-cell patch-clamp technique under similar conditions to the ones described previously[25]. In line with the previous reports[25,31,33], the application of substrate glutamate increased the amplitudes of anion leak currents (Supplementary Fig. 1a). In contrast, the WAY-213613 was able to block the anion leak conductance (Supplementary Fig. 1b). The activation of anion leak currents mediated by glutamate and inhibition of the ones by WAY-213613 were both dose-dependent and could be fitted into a Michaelis–Menten-type equation, yielding a $K_m$ of 30 ± 2.5 μM for glutamate and an apparent $K_i$ of 0.07 ± 0.03 μM for WAY-213613 (Fig. 1a, b), both of which are consistent with previous reports[26,34].

To elucidate the structural features of hEAAT2, we expressed and purified the wild-type hEAAT2 in HEK293 cells (Supplementary Fig. 1c). The cryo-EM studies were carried out in the presence of substrate glutamate and inhibitor WAY-213613, generating two 3.4 Å maps (Supplementary Figs. 2, 3 and Supplementary Table 1). hEAAT2 has a homotrimer structure resembling other EAATs (hEAAT1[19] and hEAAT3[20]) and its orthologs (hASCT1[21] and hASCT2[22–25]) or paralogs (GltPh[4,9,10,14,15] and GltTk[16,18]). Each subunit of the hEAAT2 trimer has eight transmembrane helices (TM1-TM8) and a pair of half transmembrane spiral hairpins (HP1, HP2) (Fig. 1c, d). These secondary structures assemble into a scaffold domain (TM1, 2, 4, 5 and two extracellular beta sheets insert in TM4b and TM4c) and a transport domain (TM3, 6–8 and HP1, HP2), respectively. Each of the two domains exhibits a recognizable internal double symmetry. The scaffold domain of each individual protomer connects with those of the other two protomers, which constitutes a fairly stable integration at the central part of the homotrimer. And the transport domain is separated by its own scaffold domain and distributed like an equilateral triangle (Fig. 1c). In our hEAAT2 structures, the HP1 is situated approximately parallel to the membrane plane and almost all exposed in the cytoplasm (Fig. 1d). The counterpart HP2 is embedded in the lipid bilayer and the HP2 tip is solvent-accessible from the cytoplasmic side (Fig. 1d). The hEAAT2 has a length of ~97 Å and a width of ~100 Å along normal to the membrane plane, and ~70 Å along the membrane normal (Fig. 1c, d).

Owing to the high resolution of two resolved structures of hEAAT2, many strip-shaped densities distributing around the protein were observed (Supplementary Fig. 4a, b). The most abundant lipids are located between the two adjacent subunits of hEAAT2 trimer, which is similar to the distribution of lipids found in the structures of hASCT2 (6RVX)[23] or GltPh (6 × 15)[14]. During the preparation of hEAAT2 samples, we found that CHS has a significant impact on the homogeneity and SEC profile of hEAAT2, as a similar scenario reported in hASCT2[20]. In fact, CHS resembling densities are determined to be located in the cavity formed by TM3, TM6, and TM8 at the inner lobe of the plasma membrane (Supplementary Fig. 4c, d), which was also found in the maps of hEAAT3[35] and GltPh[14] (6S3Q and 6 × 15, respectively), suggesting that cholesterol may be correlated to the assembly or function of SLC1 transporters and their homologs.

### Recognition of the glutamate by hEAAT2.
Mammalian EAATs transport L-glutamate, L-aspartate, and D-aspartate with apparent micromolar affinities[3]. In order to understand how hEAAT2 binds with glutamate, we determined the structure of hEAAT2 in complex with the glutamate (hEAAT2$^{Glu}$) at 3.4 Å resolution (Supplementary Fig. 2). In the hEAAT2$^{Glu}$ structure, the glutamate is sealed by the tips of HP1 and HP2 loops (Fig. 2a, b). Using the scaffold domain as a reference, we performed a

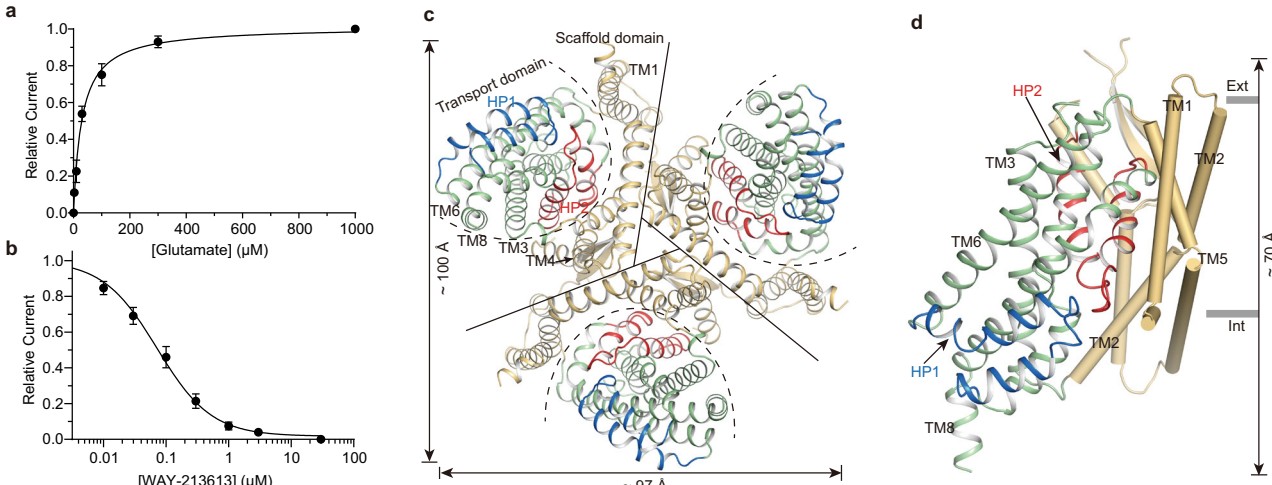

**Fig. 1 Functional characterization and overall architecture of the hEAAT2. a** Glutamate dose-response relationship for hEAAT2-associated current in the presence of 3 µM, 10 µM, 30 µM, 100 µM, 300 µM, and 1000 µM glutamate. The sample sizes ($n$) tested from low to high concentrations are 5, 5, 5, 5, 5, and 8 cells. The lines represent the best fit to a Michaelis-Menten-like equation with an average apparent $K_m$ of 30 ± 2.5 µM for glutamate. Currents were normalized to the maximal current recorded after the application of 1000 µM glutamate. **b** WAY-213613 dose-response relationship for hEAAT2-associated current in the presence of 0.01 µM, 0.03 µM, 0.1 µM, 0.3 µM, 1 µM, 3 µM, and 30 µM WAY-213613. The sample sizes ($n$) tested from low to high concentrations are 5, 5, 5, 5, 5, 5, and 10 cells. The lines represent the best fit to a Michaelis-Menten-like equation with an average apparent $K_i$ of 0.07 ± 0.03 µM for WAY-213613. Currents were normalized to the maximal current recorded after application of 30 µM WAY-213613. In figures **a** and **b**, all experiments were executed at 0 mV, and data were presented as mean values ± SD. **c** Cartoon representation of the hEAAT2 homotrimer viewed from the cytoplasm. Solid lines divide the homotrimer into three single protomers and dashed curved lines delineate the transport domain of each protomer. Each protomer consists of the scaffold domain (wheat) and the transport domain (pale green). Structural elements HP1 (blue) and HP2 (red) are highlighted. The same color scheme is adopted throughout the manuscript unless specifically indicated. **d** Structure of single protomer viewed from the membrane plane. The transport domain, and HP1 and HP2 are shown as cartoon while the scaffold domain is shown as cylinders. In figures **c** and **d**, the boundaries of the cell membrane are represented by gray lines and dimension information are labeled alongside the arrows.

structural comparison between the outward-facing hEAAT1$^{Asp}$ (PDB ID: 5LLU)[19] and the hEAAT2$^{Glu}$ structures and found that the transport domain of hEAAT2$^{Glu}$ structure is displaced by ~17 Å across the membrane towards cytoplasmic side relative to that of hEAAT1$^{Asp}$, suggesting that the hEAAT2$^{Glu}$ structure is determined in an inward-facing conformation (Supplementary Fig. 5). Such conformational change of the transport domain is also consistent with previous observations[24]. At the location of highly conserved polar residues from an unwound region of TM7 (NMDGT motif[10]), the amphipathic TM8 forms a wide range of strong interactions (mainly hydrogen bond interactions) with glutamate to stabilize the binding of the latter to the hEAAT2 (Fig. 2a). Specifically, the α-carboxyl group of the substrate glutamate forms contacts with the side chains of T479$^{TM8}$ and N482$^{TM8}$, and the main-chain N of S364$^{HP1}$ in the tip of HP1. The amino group of the substrate glutamate is coordinated only with the carboxyl group of D475$^{TM8}$. The δ-carboxyl group of the substrate glutamate interacts with the side chains of T401$^{TM7}$ and R478$^{TM8}$ (Fig. 2c, d). By comparison with eukaryotic homologs (hEAAT1[19], hEAAT3[20], hASCT2[22]) and prokaryotic homologs (Glt$_{Ph}$[14], Glt$_{Tk}$[17]), key residues involved in substrate binding are highly conserved between hEAAT2 and the above-mentioned homologs (Supplementary Fig. 6). However, R478$^{TM8}$ in the hEAAT2 is substituted by C467$^{TM8}$ at the corresponding position in the neutral amino acid transporter hASCT2 (Supplementary Fig. 6e), which contributes to the substrate selectivity of hASCT2 for neutral amino acids[20]. To validate the glutamate binding site, we mutated D475$^{TM8}$ and R478$^{TM8}$ to alanine separately (D475A$^{TM8}$ and R478A$^{TM8}$). Electrophysiological experiments showed that glutamate failed to activate anion currents of these two mutants at the concentration of glutamate up to 1 mM, as compared with that of wild-type hEAAT2 (Fig. 2e and Supplementary Fig. 7a–c). To rule out the possibility that mutation in

either residue might disrupt the activity of hEAAT2 as an anion channel, we also tested the inhibitory effects of WAY-213613 on anion currents of these two mutants. As a result, both mutants, D475A$^{TM8}$ and R478A$^{TM8}$, still could mediate an inward anion current, evidenced by an apparent outward current induced by the WAY-213613, although the apparent $K_i$ of WAY-213613 was reduced to 30.48 µM and 1.68 µM for D475A$^{TM8}$ and R478A$^{TM8}$, respectively (Fig. 2f and Supplementary Fig. 7e, f). Therefore, we speculate that the residues D475$^{TM8}$ and R478$^{TM8}$ are critical for the binding of hEAAT2 with glutamate and might also participate in the binding of this transporter with WAY-213613.

Compared with inward-facing hEAAT3 at apo state (PDB ID: 6X3F)[20], we found that the HP2 in the hEAAT2$^{Glu}$ complex remarkably shifts towards the intracellular side when glutamate is bound (Fig. 2g), which would prevent the escape of substrate. This conformation is stabilized by hydrogen bonds between HP2 and surrounding helices. In addition to directly participating in the glutamate binding, D475$^{TM8}$ can also form one hydrogen bond with the backbone nitrogen of S444$^{HP2}$ from the HP2, which contributes to the close-up of HP2 tip (Fig. 2g). Moreover, the backbone carbonyl group and the side chain of S441$^{HP2}$ interact with the side chains of S363$^{HP1}$ and A394$^{TM7}$, respectively (Fig. 2g). Taken together, three pairs of hydrogen bonds, including D475-S444, S363-S441, and S441-A394, are critical to keep HP1 and HP2 staying close to each other. Interestingly, the S441$^{HP2}$ is only present in the hEAAT2 and is substituted by a conserved glycine residue at the corresponding position of each homolog (Supplementary Fig. 8). To investigate the functional role of the S441$^{HP2}$, we substituted this residue with glycine (S441G$^{HP2}$) and performed electrophysiological recordings by applications of various concentrations of glutamate. Strikingly, we found that the S441G$^{HP2}$ mutant displays higher sensitivity to glutamate with the $K_m$ at 0.40 ± 0.03 µM (Fig. 2h and Supplementary Fig. 7d), which is

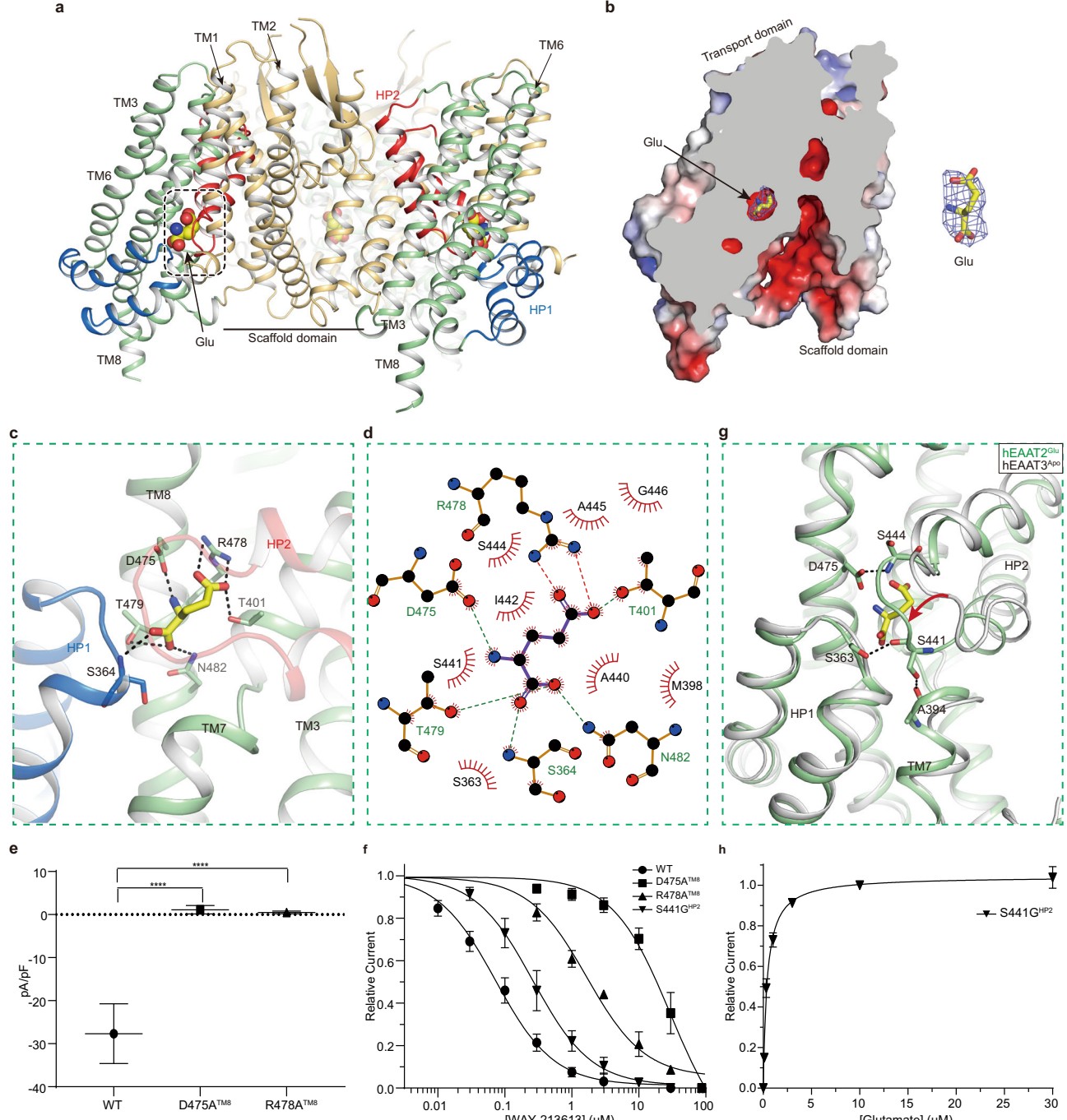

increased by ~77-fold higher than that of wild-type hEAAT2, although the S441$^{HP2}$ is not directly involved in the glutamate binding (Fig. 2c, d). This residue is exclusively present in the hEAAT2, as compared with the conserved glycine at the corresponding position in other homologs (Supplementary Fig. 8). We speculate that the unique residue S441$^{HP2}$ probably provides additional interactions to stabilize the HP2 at a sealed confirmation and prohibit glutamate release from the intracellular side. Thus, mutation in the S441$^{HP2}$ might rupture these interactions, facilitate the glutamate release, accelerate the glutamate uptake cycle, and consequently biases the hEAAT2 towards a channel with higher open probability. The apparent $K_i$ of WAY-213613 for the mutant S441G$^{HP2}$ was reduced to 0.26 μM (Fig. 2f and Supplementary Fig. 7g), which hints that this site also affects the binding of the inhibitor in a certain way.

**Antagonism of the WAY-213613**. The WAY-213613 displays high selectivity for the hEAAT2, thus it is desirable to investigate its specificity differences between hEAATs. To investigate the inhibitory mechanism of hEAAT2 by WAY-213613, we determined a complex structure of the hEAAT2 with this inhibitor at 3.4 Å resolution (Fig. 3a and Supplementary Fig. 3). The WAY-213613 is composed of three functional groups, including bromofluorophenol, aniline, and asparagine groups (Fig. 3b). This complex structure adopts an inward-facing conformation. An extra density is identified at the glutamate-binding site close to the cytoplasmic side of the membrane, which is well compatible with the inhibitor WAY-213613 (Fig. 3b). Hereafter, we termed the WAY-213613 bound hEAAT2 as hEAAT2$^W$ complex. The interactions involved in stabilizing the inhibitor WAY-213613 are composed of the hydrophobic interactions and hydrogen bonds

**Fig. 2 Glutamate binding site of the hEAAT2. a** Overall structure of the hEAAT2$^{Glu}$ homotrimer with glutamate (colored spheres) viewed from the membrane plane. **b** Slice through the molecular surface of a single protomer of hEAAT2$^{Glu}$ showing glutamate (sticks) and its EM density (blue mesh). **c** Detailed binding site for glutamate (yellow) with its key interacting residues (sticks). **d** 2D plot representing the interactions of glutamate with its surrounding residues by LigPlot$^+$. **e** Comparison of the glutamate-related current densities between the tested mutants and wild-type hEAAT2. Currents were recorded at 1000 μM glutamate and presented after Na$^+$ dependent leak current was subtracted. Sample sizes (n) tested for wild-type hEAAT2, D475A$^{TM8}$, or R478A$^{TM8}$ are 5, 4, 5 cells. One-way ANOVA was followed by the Bonferroni's post hoc test, ****$P < 0.0001$. **f** WAY-213613 dose-response relationships for the D475A$^{TM8}$, R478A$^{TM8}$, or S441G$^{HP2}$ mediated currents. WAY-213613 was varied at the following concentrations: 0.3 μM, 1 μM, 3 μM, 10 μM, 30 μM, and 100 μM for D475A$^{TM8}$ and R478A$^{TM8}$; 0.03 μM, 0.1 μM, 0.3 μM, 1 μM, 3 μM, 10 μM, and 30 μM for S441G$^{HP2}$. Currents were normalized to the maximal current recorded after application of 30 μM or 100 μM WAY-213613. Sample sizes (n) tested from low to high concentrations are listed as follows: $n = 4, 5, 5, 5, 5$, and 9 cells for D475A$^{TM8}$; $n = 5, 5, 5, 4, 5$, and 12 cells for R478A$^{TM8}$; $n = 4, 4, 5, 5, 4$, and 7 cells for S441G$^{HP2}$. The lines represent the best fits to a Michaelis-Menten-like equation. The data of wild-type hEAAT2 from Fig. 1b was compared with those of the tested mutants. **g** Structural comparison of the transport domains between hEAAT2$^{Glu}$ (pale green) and hEAAT3$^{Apo}$ (PDB ID: 6X3F, gray). **h** Glutamate dose-response relationship for the S441G$^{HP2}$ mediated currents at the varied concentrations of 0.1 μM, 0.3 μM, 1 μM, 3 μM, 10 μM, and 30 μM. The lines represent the best fit to a Michaelis–Menten-like equation with an average apparent $K_m$ of 0.40 ± 0.03 μM. Currents were normalized to the maximal current recorded after the application of 30 μM glutamate. Sample sizes (n) tested from low to high concentrations are 4, 5, 5, 5, 12, and 5 cells. In the figures, **e**, **f** and **h**, all experiments were executed at 0 mV and the data were presented as mean values ± SD.

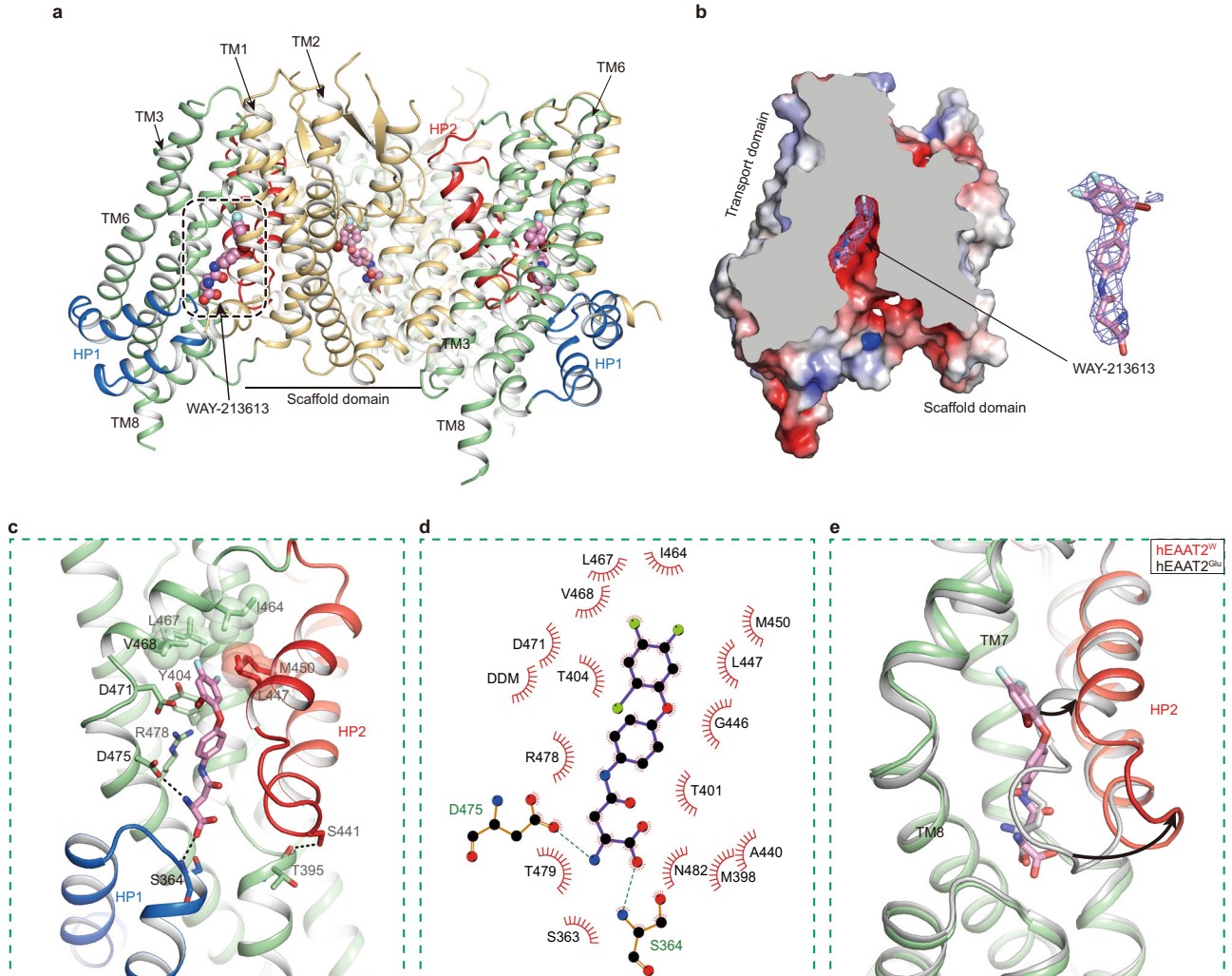

**Fig. 3 WAY-213613 binding site of the hEAAT2. a** Overall structure of hEAAT2$^W$ viewed from the membrane plane. WAY-213613 is shown as spheres. **b** Slice through the molecular surface of single protomer of hEAAT2$^W$. The WAY-213613 is displayed in sticks, and its corresponding EM density is shown in blue mesh. **c** Detailed binding site for WAY-213613 in hEAAT2$^W$. The inhibitor WAY-213613 (pink) and the key interaction residues are shown as sticks; the hydrophobic residues of the binding pocket are shown as half-transparent spheres. **d** 2D plot representing the interactions of WAY-213613 with its surrounding residues by LigPlot$^+$. **e** Conformational changes of the glutamate binding site between hEAAT2$^W$ (pale green) and hEAAT2$^{Glu}$ (gray). The HP2 of hEAAT2$^W$ is highlighted in red and black arrows indicate the shift of the HP2.

between the inhibitor and its surrounding residues (Fig. 3c, d). More specifically, the α-carboxyl group of WAY-213613 forms contacts with the backbone nitrogen and side-chain hydroxyl group of S364$^{HP1}$. The amino group of WAY-213613 is coordinated with the side-chain carboxyl group of D475$^{TM8}$ (Fig. 3c). Moreover, R478$^{TM8}$ not only forms a salt bridge with D475$^{TM8}$, but also forms two cation-π interactions with Y404$^{TM7}$ and the aniline group of WAY-213613 (Fig. 3c). These structural observations are also supported by the functional analysis that the mutants D475A$^{TM8}$ and R478A$^{TM8}$ exhibited significantly reduced binding affinities with the WAY-213613 as compared with that of wild-type hEAAT2 (Fig. 2f). Furthermore, the bromofluorophenol group is wrapped in a hydrophobic cavity formed by the hydrophobic residues I464$^{TM8}$, L467$^{TM8}$, V468$^{TM8}$, M450$^{HP2}$, and L447$^{HP2}$ (Fig. 3c). The bromofluorophenol group is also stabilized by forming π−π stack interaction with Y404$^{TM7}$ in a T-shape configuration (Fig. 3c). In the cryo-EM map of hEAAT2$^W$ complex, a DDM-like molecule was found to be sandwiched between TM1 and the unwound helix of TM8. Its hydrophobic tail points towards the extracellular side. The head group is positioned proximal to the bromofluorophenol group and form hydrophobic interactions to stabilize the WAY-213613 binding (Supplementary Fig. 4b, e). Therefore, we speculate that a lipid molecule in the membrane environment may occupy the binding position of the DDM and participate in the inhibitory effects of glutamate transport by the WAY-213613.

We also superimposed the structures of the hEAAT2$^{Glu}$ and hEAAT2$^W$ (Fig. 3e). Since a part of the WAY-213613 is derived from asparagine, it was not surprising that the inhibitor forms similar hydrogen bond interactions in parallel with the situation of glutamate binding. The α-carboxyl and α-amino groups of WAY-213613 or glutamate are coordinated by the same cluster of residues from TM8 and the HP1 tip, such as D475$^{TM8}$ and S364$^{HP1}$ (Figs. 2c, 3c). However, upon WAY-213613 binding, the hydrophobic bromofluorophenol group of the inhibitor interacts with hydrophobic residues from the HP2, such as M450$^{HP2}$ and L447$^{HP2}$, and consequently drives the HP2 away from the HP1 tip and opens the HP2 tip. The HP2b spiral undergoes an outward-facing rotation with an angle of about 22°, and the HP2 tip has a distance change at ~8.0 Å (Fig. 3e). Just owing to the movement of the HP2, the residues S441$^{HP2}$ and S444$^{HP2}$ at the HP2 tip are unable to form hydrogen bonds with the HP1 tip and TM8, respectively, which is present in the glutamate bound hEAAT2 structure (Fig. 2g). Instead, the hydroxy group S441$^{HP2}$ forms hydrogen bond with the carbonyl group of T395$^{TM7}$ and thus stabilized the WAY-213613 bound state (Fig. 3c), in line with the functional analysis mentioned above that S441G$^{HP2}$ mutation decreased the sensitivity of hEAAT2 to WAY-213613 (Fig. 2f). As the residues participating in WAY-213613 binding are from HP1, HP2, and TM8 elements, this interaction network mediated by WAY-213613 could consequently stabilize the hEAAT2 in an inward-facing conformation state. Moreover, such an open conformation of the HP2 stabilized by WAY-213613 hampers the conformational change, a key process for hEAAT2 activity.

**Specificity of the WAY-213613.** To understand how the WAY-213613 selectively blocks the hEAAT2, we superimposed the structure of inward-facing hEAAT3$^{Na}$ (PDB ID: 6X2L)[20] with that of the hEAAT2$^W$ (Fig. 4a–c). We found that the asparagine group of WAY-213613 is fairly compatible with the structure of hEAAT3. However, some residues adjacent to the bromofluorophenol groups are substituted in hEAAT3. In particular, the residues V468$^{TM8}$, L467$^{TM8}$, and I464$^{TM8}$ from TM8 of hEAAT2

are substituted by I437$^{TM8}$, I436$^{TM8}$, and V433$^{TM8}$ in hEAAT3, respectively (Fig. 4c and Supplementary Fig. 8). Consequently, these substitutions in hEAAT3 lead to the shrinkage of the binding pocket of the WAY-213613 in hEAAT3, implying the mechanism why hEAAT3 cannot exhibit the same high affinity to the inhibitor WAY-213613 as that of hEAAT2 (Fig. 4a, b). To validate the above hypotheses, we designed three mutations, including I464V$^{TM8}$, L467I$^{TM8}$, and V468I$^{TM8}$. The electrophysiological experiments indicate that mutations do not affect glutamate binding. For these three mutants, the application of glutamate in the external solution can significantly activate anion current with similar efficacy as that of wild-type hEAAT2 (Fig. 4d and Supplementary Fig. 9a–c). However, these mutations lead to a significant decrease in sensitivity to WAY-213613, with $K_i$ increased to ~0.17 μM, ~0.46 μM, and ~0.20 μM for I464V$^{TM8}$, L467I$^{TM8}$, and V468I$^{TM8}$ mutants, respectively (Fig. 4e and Supplementary Fig. 9d–f), supporting our speculations that the I464$^{TM8}$, L464$^{TM8}$, and V468$^{TM8}$ residues are crucial for the binding specificity of hEAAT2 with WAY-213613.

So far, a series of structures of hEAAT2 homologs have been determined in the presence of the different kinds of inhibitors, including competitive inhibitors TBOA[10,14,18], TFB-TBOA[14,19,35], trans and cis isomers of *p*-OMe-azo-TBOA[36], *Lc*-BPE[25], and allosteric inhibitors UCPH-101[19]. Complex structures with inward-facing conformation were adopted for further comparison with hEAAT2$^W$ (Supplementary Fig. 10a), comprising Glt$_{Ph}$$^{TFB-TBOA}$ (PDB ID: 6×14)[14], Glt$_{Ph}$$^{TBOA}$ (PDB ID: 6×16)[14], and hEAAT3$^{TFB-TBOA}$ (PDB ID: 6S3Q)[35]. In the structures of Glt$_{Ph}$$^{TBOA}$ and hEAAT3$^{TFB-TBOA}$, the inhibitors insert into the two helices of the HP2 to separate HP2a from HP2b, and thus block the conformational change of the transporter (Supplementary Fig. 10b, c). However, the orientation of WAY-213613 in the hEAAT2$^W$ complex is significantly different from those of the inhibitors in the Glt$_{Ph}$$^{TBOA}$ and hEAAT3$^{TFB-TBOA}$ complexes (Supplementary Fig. 10a–c). The bromofluorophenol group of WAY-213613 is located in the "deep hydrophobic cavity" formed by TM8, TM7b, and HP2b (Fig. 3c and Supplementary Fig. 10a). The Glt$_{Ph}$$^{TFB-TBOA}$ complex displays a distinct binding site from that of the hEAAT3$^{TFB-TBOA}$ complex (Supplementary Fig. 10c, d). In contrast, the Glt$_{Ph}$$^{TFB-TBOA}$ complex seems to display a similar binding site from that of the hEAAT2$^W$ complex (Supplementary Fig. 10a, d). However, the TFB-TBOA is unable to approach to the "deep cavity" of the WAY-213613 in the hEAAT2 (Supplementary Fig. 10e), The above results reveal that the WAY-213613 is bound at a previously unidentified cavity.

## Discussion

hEAAT2 is the most abundantly expressed excitatory amino acid transporter in astrocytes, which is obligate for the uptake of glutamate from the synaptic cleft into astrocytes to prevent neuronal cytotoxicity[3]. Here, we report the cryo-EM structure of hEAAT2$^{Glu}$ in complex with glutamate, which reveals the molecular details of how hEAAT2 recognizes the glutamate. By comparison with eukaryotic homologs (hEAAT1[19], hEAAT3[20], hASCT2[22]) and prokaryotic homologs (Glt$_{Ph}$[14], Glt$_{Tk}$[17]) (Supplementary Fig. 6), the key interaction residues involved in substrate binding are highly conserved between hEAAT2 and the above-mentioned homologs. Meanwhile, we found that a unique residue S441$^{HP2}$ is exclusively present in hEAAT2 and it is substituted by the conserved glycine at the corresponding position of other homologs (Supplementary Fig. 8). Electrophysiological experiments showed that the affinity of S441G$^{HP2}$ to glutamate can be significantly increased as compared with that of wild-type hEAAT2 (Fig. 2h). Notably, the S441$^{HP2}$ is located at the HP2 tip and does not directly take part in the interaction with glutamate

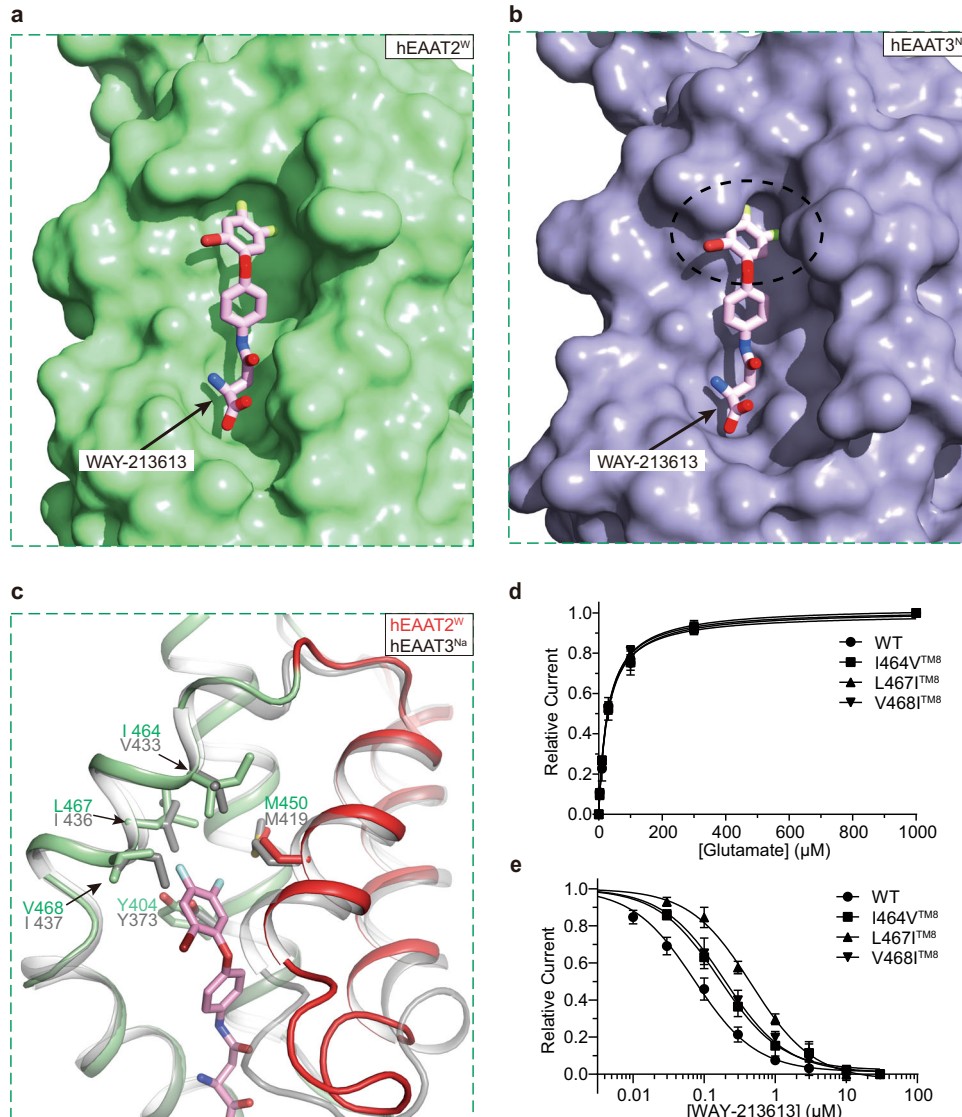

**Fig. 4 Structural basis for the selectivity of WAY-213613. a** Surface representation of the WAY-213613 pocket in the hEAAT2$^W$ (pale green). The WAY-213613 is displayed in sticks (pink). **b** Docking of WAY-213613 into the hEAAT3$^{Na}$ (PDB ID: 6X2L). The surface representation of hEAAT3$^{Na}$ (blue) indicates a shrinkage of the WAY-213613 binding pocket in hEAAT3. The WAY-213613 is displayed in sticks (pink). **c** Structural comparison of the transport domain of hEAAT2$^W$ (pale green) and hEAAT3$^{Na}$ (PDB ID: 6X2L, gray). WAY-213613 and hydrophobic residues in its binding pocket are displayed in sticks. The HP2 of hEAAT2$^W$ is colored in red. **d** Glutamate dose-response relationships for the I464V$^{TM8}$, L467I$^{TM8}$, or V468I$^{TM8}$ mediated currents. The dose-response relationships for the tested mutants were analyzed at the varied concentrations of glutamate indicated in Fig. 1a. Sample sizes (*n*) tested from low to high concentrations are listed as follows: $n = 5, 5, 5, 5, 5$, and 5 cells for the mutant I464V$^{TM8}$; $n = 4, 4, 5, 5, 5$, and 5 cells for the mutant L467I$^{TM8}$; $n = 5, 5, 5, 5, 4$, and 6 cells for the mutant V468I$^{TM8}$. **e** WAY-213613 dose-response relationships for the I464V$^{TM8}$, L467I$^{TM8}$, or V468I$^{TM8}$ mediated currents. The dose-response relationships for the hEAAT2 mutants as WAY-213613 were varied at the indicated concentrations in Fig. 1b. The lines represent the best fits to a Michaelis-Menten-like equation with apparent $K_i$ of 0.166 µM, 0.464 µM, and 0.196 µM, respectively. Currents were normalized to the maximal current recorded after application of 30 µM WAY-213613. Sample sizes (*n*) tested from low to high concentrations are: $n = 5, 5, 5, 4, 4, 5$, and 10 cells for I464V$^{TM8}$; $n = 5, 5, 5, 5, 5, 4$, and 11 cells for L467I$^{TM8}$; $n = 5, 5, 5, 5, 5, 5$, and 11 cells for V468I$^{TM8}$. The data of wild-type hEAAT2 from Fig. 1b was compared with those of the tested mutants. In figs. **d** and **e**, all experiments were executed at 0 mV and data were presented as mean values ± SD.

(Fig. 2g). Thus, we speculate that S441$^{HP2}$ may affect the rate of transport glutamate. Further experiments will be required to clarify the functional roles of S441$^{HP2}$ in hEAAT2.

In this study, we also resolved the structure of hEAAT2 in complex with the inhibitor WAY-213613, which clearly elucidates the binding pocket of hEAAT2 for WAY-213613. The hEAAT2$^W$ complex is stabilized at the inward-facing conformational state. The α-carboxyl and α-amino and the aniline groups of WAY-213613 form contacts with residues S364$^{HP1}$, D475$^{TM8}$, and

R478$^{TM8}$ and share an overlapped binding site with the substrate glutamate, in line with a notion that WAY-213613 is a competitive inhibitor. In the hEAAT2$^W$ complex, the HP2 undergoes a remarkably conformational change and rotates away from the HP1, thus creating enough space for the WAY-213613 binding. Interestingly, the S441$^{HP2}$ was determined to form a hydrogen bond with T395$^{TM7}$ and consequently stabilize the HP2 in an open conformation, which turns out that this interaction is critical for the WAY-213613 binding (Figs. 3c, 2f). Meanwhile, the

bromofluorophenol group of WAY-213613 is wrapped by some hydrophobic residues from TM8, TM7b, and HP2b, such as I464$^{TM8}$, L467$^{TM8}$, V468$^{TM8}$, M450$^{HP2}$, and L447$^{HP2}$ (Fig. 3c). Based on the structural comparison and sequence alignment, the residues I464$^{TM8}$, L467$^{TM8}$, and V468$^{TM8}$ from TM8 in hEAAT2 are varied as compared with the corresponding residues of other hEAATs (Fig. 4c). The electrophysiological experiments demonstrate that the above three residues are critical for the inhibition of hEAAT2 by WAY-213613 with high potency and mutation in each of the three residues substantially reduces the inhibitory efficiency of hEAAT2 by WAY-213613 (Fig. 4e). Considering the I464-L467-V468 cluster in hEAAT2 is substituted by Val-Leu-Ile in hEAAT3 or Ile-Ile-Val in hEAAT1/hEAAT4/hEAAT5, respectively, we speculate that the I464-L467-V468 cluster from TM8 acts as a key structural determinant for the selective inhibition of hEAAT2 by WAY-213613.

## Methods

**EAAT2 expression and purification**. The full-length hEAAT2 (UniProtKB accession: P43004) gene was cloned from an HEK293 cDNA, and constructed into a modified pEG BacMam vector that has an N-terminal Strep· Tag II followed by a superfolder GFP (sfGFP) and a PreScission Protease (PPase) recognition site. The Bac-to-Bac baculovirus expression system (Invitrogen) was used to express hEAAT2 recombinant proteins in HEK293F cells. Briefly, the plasmid pEG-strep-GFP-hEAAT2 was transformed into *Escherichia coli* DH10bac cells to acquire a bacmid plasmid containing the hEAAT2 gene, in accordance with the manufacturer's instructions (Bac-to-Bac; Invitrogen). Then the baculovirus was obtained in Sf9 cells (*Spodoptera frugiperda*-9 cells), and P2 viruses were collected after 72 h of amplification. HEK 293F cells were cultured to a density of $2.5 \times 10^6$ cells/ml at 37 °C, and the P2 viruses were added at a ratio of 1% (v/v) to initiate transfection and simultaneously supplemented with 1 mM glutamate (Sigma-Aldrich) or 5 μM WAY-213613 (MedChemExpress), and 1% (v/v) FBS (fetal bovine serum), then incubated in a shaker at 37 °C with 5% CO$_2$. After the 8–12 h cultivation, 10 mM sodium butyrate was added and the cells continued to be incubated for 48 h followed by centrifugation at 2500 × g for 5 min at 4 °C for cell collection.

The cells pellets were resuspended in buffer A (20 mM HEPES, 150 mM NaCl, pH 7.5, 1 mM EDTA, 1 mM glutamate, or 5 μM WAY-213613) supplemented with a 1:1000 dilution of mammalian protease inhibitor cocktail (Sigma-Aldrich) or flash-frozen by liquid nitrogen and stored at −80 °C for further use. The resuspended cells were disrupted in a Dounce homogenizer. Cell debris was removed by centrifugation at 10,000 × g for 10 min at 4 °C, and the supernatant was ultra-centrifuged at 100,000 × g for 0.5 h at 4 °C. The crude membrane pellets were resuspended in buffer A supplemented with protease inhibitor cocktail and homogenized. 1% (w/v) n-Dodecyl-β-D-Maltopyranoside (DDM; Anatrace) and 0.2% (w/v) cholesteryl hemisuccinate (CHS; Sigma-Aldrich) were added into the homogeneous solution, followed by 2 h solubilization with gentle agitation at 4 °C. After ultracentrifugation at 100,000 × g for 0.5 h at 4 °C, insoluble debris was removed and the solubilized material was filtered through a 0.22 μM filter (Merck Millipore). Subsequently, the filtrate slowly passed through a Streptactin Beads (Smart-Lifesciences) column pre-equilibrated with the buffer B (buffer A supplemented with 0.025% (w/v) DDM and 0.005% (w/v) CHS) at a rate of ~0.2 ml/min. The column was washed with 10 column volumes (CV) of buffer B to remove unbound materials, and the protein was eluted with four-column volumes of buffer C (buffer B supplemented with 5 mM D-desthiobiotin). N-terminal Strep·Tag II and GFP were digested by incubating with an appropriate ratio of homemade PreScission protease at 4 °C for 3 h. hEAAT2 sample was concentrated to 1 ml using a 100 kDa cut-off concentrator (Merck Millipore) and further purified by size exclusion chromatography using Superose 6 Increase 10/300 GL (GE Healthcare) pre-equilibrated with the buffer B. Finally, the fractions containing purified protein were collected and immediately concentrated for cryo-EM grid preparation.

**Cryo-EM sample preparation and data acquisition**. To prepare grids for cryo-EM imaging, the freshly purified protein was concentrated and supplemented with 2 mM glutamate or 200 μM WAY-213613 on ice and incubated for 30 min. In all, 2.5 μl of 12.6 mg/ml hEAAT2$^{Glu}$ complex or 10 mg/ml hEAAT2$^W$ complex was applied to holey-carbon cryo-EM grids (Quantifoil Cu R1.2/1.3, 300 mesh), which were glow-discharged for 60 s in H$_2$O$_2$ condition by the Solarus plasma cleaner (Gatan). Grids were blotted either for 2.5 s at 4 °C and 100% humidity in Vitrobot Mark IV (Thermo Fisher Scientific), subsequently vitrified by plunge-frozen into liquid ethane and stored in liquid nitrogen. Cryo-EM data were acquired using a 300 kV Titan Krios transmission electron microscope (Thermo Fisher Scientific) equipped with a Gatan K2 Summit direct electron detector (Gatan) positioned after a GIF quantum energy filter. With slit of energy filter set to 20 eV, SerialEM[37] was used for robot collection of movie stacks in super-resolution counting mode at ×130,000 magnification (1.04 Å pixel size) with a nominal defocus range of

1.5–2.5 μm. Movie stacks had a total dose of about 50 e/Å$^2$ distributed over 60 frames with the dose rate range of 8.5–9.0 e$^-$/Å$^2$/s.

**Data processing**. For the dataset of hEAAT2$^{Glu}$ complex, a total of 2256 dose-fractionated movies were collected, beam-induced motion was corrected using MotionCor2[38] and the contrast transfer function (CTF) estimation was determined by GCTF[39]. 148 images which revealing poor CTF estimation or showing contamination with ice were discarded, thus resulting in 2108 images which were selected for succeeding analysis with RELION-3.1[40] unless specifically mentioned. A total of 1,509,287 particles were automatically picked using Gautomatch, and initial references were calculated using Ab-initio Reconstruction in cryoSPARC[41]. Two rounds of guided multi-reference 3D classification were performed against several maps biased and resulting good references in which C3 symmetry was imposed. The resulting class 4 (41.8%) in the first round of 3D classification and class 4 (82.8%) in the second round of 3D classification displayed clearly-resolved transmembrane helices, and was submitted for the following 3D refinement, which yielded a 3.9 Å resolution map. To further improve the quality of hEAAT2 map, 3D classifications without alignment was performed with using a tight mask and C3 symmetry imposed. Particles composing the best one class of 6 classes (15.0%) were picked for further CTF-refinement and 3D-auto refinement using C3 symmetry and postprocessing with tight mask, and improved the final map to 3.4 Å resolution according to the GSFSC criterion[42] reported.

For the dataset of hEAAT2$^W$ complex, a total of 465 dose-fractionated movies were collected, beam-induced motion was corrected using MotionCor2[38] and the CTF estimation was determined by GCTF[39]. A total of 233,008 particles were selected from the 420 micrographs using Template picker in cryoSPARC[41]. The extracted particles were subjected to two rounds of 2D classification. The good 2D class averages were selected and were subjected to 3D Heterogeneous Refinement with C3 symmetry imposed. The resulting good 3D class with 98,685 particles was subjected to 3D Non-uniform Refinement to yield a map at 3.7 Å resolution. The 98,685 particles were re-extracted in RELION-3.1[40], followed by 3D auto-refinement, postprocessing, and polishing to further improve the particle quality, and then the polished particles were re-imported to cryoSPARC. After 3D heterogeneous refinement with C3 symmetry imposed, 98,536 particles from the best class were subjected to non-uniform refinement. Ultimately, a map at 3.4 Å resolution was obtained.

**Model building and refinement**. For model building of hEAAT2$^{Glu}$ complex, a previously released cryo-EM structures of hEAAT3 (PDB ID: 6S3Q)[35] removing its non-protein components was used as reference model docking into the density map using Chimera[43], followed by manually adjustment in COOT[44]. This high-resolution map can visibly assign the protein sequence and confidently model the majority of residues (37–148, 162–195, and 229–507). An orphaned solid density was found in each protomer at the spatial center lidding by the top of HP2 and HP1, which was deemed to be a density for 2 mM glutamate preexisting when grids were prepared and analogous substrate aspartate or glutamine was bound to a similar site in hEAAT2 homologs. Glutamate molecule was manually docked into the density using COOT. Restraints for PC1 (1,2-Diacyl-sn-glycero-3-phos-phocholine) and CHS were derived by eLBOW[45] in Phenix. Here PC1 was used as a model lipid with removing acyl chains or ethanolamine heads to fit the corresponding densities. Subsequently, the model was subjected to manual adjustment in COOT and refined against the map utilizing Phenix with rounds of real-space refinement[46] with default parameters. MolProbity[47] was used for validation. The final model of hEAAT2 gained good geometry (Supplementary Table 1). Fourier Shell Correlation (FSC) curves were calculated between the final refined unmasked model and masked map, as well as the cross-validation of refined model versus the map of hEAAT2 (Supplementary Fig. 2 and Supplementary Table 1). The figures were prepared using UCSF ChimeraX[48] and PyMOL[49] or LigPlot$^+$[50].

**Electrophysiology**. Human embryonic kidney 293T (HEK293T) cells were cultured in Dulbecco's Modified Eagle Medium (DMEM, Gibco) media supplemented with 15% (v/v) fetal bovine serum (FBS, PAN-Biotech) at 37 °C with 5% CO$_2$. HEK293T cells were grown in the culture dishes ($d = 3.5$ cm) (Thermo Fisher Scientific) for 24 h and then 1 μg hEAAT2 wild-type and mutant plasmids for per dishes were used to transiently transfect HEK293T using 0.7 μg Lipofectamine 2000 Reagent (Thermo Fisher Scientific). Cells were analyzed using electrophysiological experiments between 24 and 40 h after transfection at room temperature (21–25 °C).

Electrophysiological experiments were performed as described previously with a minor modification[25]. External buffer contained 140 mM NaCl, 2 mM MgCl$_2$, 2 mM CaCl$_2$, and 10 mM HEPES (pH = 7.4 with NaOH and osmolarity of ~310 mosmol L$^{-1}$) and while internal pipette solution comprised of 130 mM NaSCN, 2 mM MgCl$_2$, 10 mM EGTA, 10 mM HEPES (pH = 7.4 with NaOH and osmolarity of ~310 mosmol L$^{-1}$) and 10 mM glutamate. 50 mM WAY-213613 stock solutions were prepared in dimethyl sulfoxide (DMSO), followed by dilutions to working concentrations with the above-mentioned external buffer. The maximal 0.5% DMSO was used in our experiment and it did not affect electrophysiological results in the control cells. Whole cell patch-clamp experiment recordings were made from isolated GFP-positive HEK293T cells (hEAAT2 wild type and mutants).

In all, 3–6 MΩ was used to fire a polished pipette (Sutter Instrument). Because relatively small current and compensation had no effect on the magnitude of observed currents, series resistance was not compensated in these experiments. Cells were immersed in the above-mentioned external buffer supplemented with a specified concentration of glutamate or WAY-213613. Glutamate-induced anion current, Na$^+$-dependent anion leak current and glutamate transport current or WAY-213613 inhibited anion leak currents were recorded using an EPC-10 amplifier (HEKA Electronic) at a 20 kHz sample rate with low pass filter at 5 kHz. All experiments were executed at 0 mV. After the cell current was stabilized, glutamate or WAY-213613 was sustained at the indicated concentrations from external dosing for at least 6 s. Subsequently, the cell was washed away with external buffer to start a new round of applications or end the experiment. The plateau currents were used for the following calculations. For analysis, the data was acquired by PatchMaster program (HEKA Electronic) at 5 s after recording. Nonlinear dose–response relationships were fitted into a Michaelis–Menten-like equation shown as below to obtain the apparent $K_m$ and $K_i$ values in the presence of glutamate or WAY-213613.

$$\frac{I}{I_{max}} = \frac{K_{m/i}}{[C] + K_{m/i}} \quad (1)$$

In this equation, $I$ represent the current at different glutamate or inhibitor concentrations, and $I_{max}$ represents the maximal current at the saturating concentration of glutamate or inhibitor. $[C]$ represents the concentration of glutamate or inhibitor. $K_{m/i}$ represents the apparent constant.

Statistical significance was assessed using one-way ANOVA followed by the Bonferroni's post-hoc test detailedly described as in the figure legends.

**Reporting summary**. Further information on research design is available in the Nature Research Reporting Summary linked to this article.

## Data availability
The three-dimensional cryo-EM density maps of the hEAAT2$^{Glu}$ and hEAAT2$^W$ have been deposited in the EM Database under the accession codes EMD-33407 and EMD-33408, respectively, and the coordinates for the structures have been deposited in protein data bank under accession codes 7XR4 and 7XR6, respectively. Source data are provided as a Source Data file. Source data are provided with this paper.

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

## Acknowledgements

We thank B. Zhu and other staff members at the Center for Biological Imaging (CBI), Core Facilities for Protein Science at the Institute of Biophysics, Chinese Academy of Science (IBP, CAS) for the support in cryo-EM data collection; We thank Bei Yang and Yan Wu for their research assistant service. This work is funded by Chinese National Programs for Brain Science and Brain-like Intelligence Technology (Grant No. 2022ZD0205800 to Y.Z.), Chinese Academy of Sciences Strategic Priority Research Program (Grant No. XDB37030304 to Y.Z.), National Key Research and Development Program of China (Grant No. 2021YFA1301501 to Y.Z.), the National Natural Science Foundation of China (Grant No. 92157102 to Y.Z. and Grant No. 32070031, 31770051 to J.J.), the National Laboratory of Biomacromolecules, Institute of Biophysics, Chinese Academy of Sciences (Grant No. 2021kf10 and 2022kf09), Chinese National Programs for Brain Science and Brain-like Intelligence Technology (Grant No. 2021ZD0202102 to Z.H.), and the National Natural Science Foundation of China (Grant No. 81371432 to Z.H.).

## Author contributions

Y.Z. conceived the project and supervised the research. Z.Z. prepared sample for cryo-EM study. H.C. and Z.Y. collected cryo-EM data and calculated the EM maps. Z.Z. and Z.Y. built and refined the atomic model. H.L., Y.D., and H.Z. help for purified of protein sample. Y.Z., J.J., and Z.Z. analyzed the structure. Z.H. and Y.Z. designed and Z.G. performed the electrophysiological experiments. Z.Z., H.C., and Z.Y. contributed to initial draft of the manuscript. Y.Z. and J.J. edited the manuscript with input from all authors in the final version.

## Competing interests

The authors declare no competing interests.
