## [Peer Review File · Nature Communications]

Structural basis of ligand binding modes of human EAAT2REVIEWER COMMENTS

Reviewer #1 (Remarks to the Author):

Zhang et. al. report human EAAT2 structures determined in glutamate and inhibitors bound states. These new structures are important, but the manuscript can be considerably improved with functional data to validate binding sites for glutamate and the inhibitor.

Minor comments:

1. The manuscript appears to be cursorily written and can be improved. For example, in line 50, hASCT1 and hASCT2 do not transport glutamate.
2. The authors may also want to compare EAAT2/glutamate with other substrate bound EAAT structures.
3. Since both EAAT/glutamate and EAAT/Asp structures are available, can these structures explain why EAAT prefers Glu over Asp (or lack of selectivity).
4. In Figure 3d, Leu447 should be Leu467?

Reviewer #2 (Remarks to the Author):

SUMMARY

In this manuscript, Zhang et al. solve the structures of human EAAT2 in complex with its substrate glutamate and inhibitor called WAY-213613 using cryo-EM. The good resolution of these structures allows for both the substrate and inhibitor to be placed in the map and consequently the mechanism of inhibition by WAY-213613 to be delineated as competitive antagonism.

MAJOR COMMENTS

1. It is mentioned that inhibitors of EAAT2 may be an important therapeutic target, but the discussion needs more depth. It seems that the authors are claiming that inhibition of EAAT2 may be clinically

however this reviewer questions the usefulness of EAAT2 inhibitors clinically given that deficiencies in EAAT2 cause disease. The authors point to Ref 11 for evidence that EAAT2 inhibitors may be clinically useful however this is actually not mentioned in the paper referred to at all.

2. While the structures provided by Zhang et al. provide significant insight as to the mechanism of inhibition by WAY-213613, for this paper to be complete, additional experiments are required that delineate a structure activity relationship. Functional experiments investigating the residues that coordinate WAY-213613 need to be conducted. For example, mutating these residues to the equivalent residues in other EAATs that have a lower affinity for WAY-213613 will help pinpoint exactly what residues are critical for the specific inhibition of EAAT2.

3. The cryoEM maps provided show clear density for glutamate and WAY-213613. Interestingly, they also show density for a DDM molecule in the glutamate binding site in the WAY-213613-bound structure. This reviewer feels that this is a very interesting observation that warrants discussion in the manuscript. Could it be that inhibition by WAY-213613 causes other molecules such as detergents – or more physiologically relevant – lipids to interact with residues in the glutamate binding site and also contributing to inhibition of transport?

4. This manuscript lacks any discussion of the results or a conclusion – these are essential aspects to scientific literature and without them this manuscript is incomplete. This reviewer feels especially strongly about this given that inhibition of glutamate transporters therapeutically is somewhat controversial and warrants, at the very least, further thought and discussion.

MINOR COMMENTS

- Line 25: “acts” should be “serves”.
- Line 26: Remove second instance of “the”.
- Line 27: It should be noted that excessive concentrations are neurotoxic.
- Line 32: “Buffering glutamate” doesn’t really make sense. Should be replaced with “transporting glutamates back up into the pre-synapse or astrocytes”.
- Lines 39-40: As noted above in major comments, more detail is warranted to explain how EAAT2 may be an important therapeutic target.
- Line 41: Ph for GltPh should be in subscript.
- Line 47: The substrate binding site is described as being “close to HP1 and HP2”. There are several substrate-bound structures of glutamate transporters and given that this is a key finding of this paper, what is already known about the substrate binding site should not be glazed over. More detail is required.
- Line 52: What is the percentage identity between human EAAT2 and other human EAATs? How much does it vary in functionally important regions?

- Line 53: It would be useful at this point of the paper to know the affinities of WAY-213613 for different human EAATs rather than at line 157.
- Supplementary Figures 2 and 3: It is difficult to see the densities. Please change the colors. Also, the densities for substrates and inhibitors should be shown in these figures.
- Lines 67-68: References are missing for the structures referred to.
- Line 76: What do the authors mean by “HP1 is attached to the inner membrane”? This reviewer is not familiar with the concept of a protein attaching to the membrane.
- Supplementary Figure 4 and Lines 86-88: While it has been described that CHS-resembling densities could be identified, this is not clearly demonstrated in Supplementary Figure 4. The authors should show how CHS fits into the density with more detail if this is their claim.
- Supplementary Figure 4: The colors for the hairpins are only shown in one of three protomers. Either say this in the legend or show it in all three protomers. Also, HP2 is listed as being blue when really it looks purple. Furthermore, WAY-213613 is said to be colored in pink, when it is actually red, and this can be very confusing with the pink-magenta used to color HP2 in this panel. Please use more distinct and consistent colors for these figures. Labels may also be helpful.
- Lines 93-95: This sentence seems out of place and is unnecessary at this point in the manuscript.
- Figure 2: I see beta sheets in the structure, but no beta sheets have been mentioned. Please explain.
- Figure 2: Density for glutamate should be shown in a close-up view.
- Lines 101-107: How does this substrate binding site compare to other substrate-bound structures? This point warrants discussion and prior work should be acknowledged here.
- Line 109-110: “Upon the glutamate binding” should be replaced with “when glutamate is bound” because here, we are talking about the release of substrate.
- Lines 115-118: Is S441 important for substrate transport? This warrants functional investigation, especially since it is not conserved amongst the different homologues. Why are these interactions of S441 significant?
- Lines 124-125: “WAY-213613 has no effect on ionotropic and metabotropic glutamate receptors and thus it’s a potential tool for elucidating the function of hEAAT2.” We already know the function of EAAT2; this is a vague sentence and does not tell the reader why they should care about understanding how WAY-213613 binds to EAAT2. Careful thought, consideration and explanation is warranted here.
- Line 128 and Figure 3B: A more zoomed-in figure of WAY-213613 in the density map should be shown.
- Figures 2D and 3D are mirror images of each other. Please amend for ease of interpretation.
- In several figures, two structures are overlaid with one structure shown in gray and the other shown in green as detailed by the figure legend. In these figures, HP2 is colored in red, but it is not detailed in the figure legend which structure this red (HP2) belongs to. Please clarify.
- Supplementary Figure 6: Please put PDB IDs into figure legends.

- Lines 168-181: There are also structures of TBOA bound to GltPh in outward-facing structures. These should also be discussed; how do they fit into this story?

Reviewer #3 (Remarks to the Author):

EAATS play a critical role in maintaining low glutamate concentrations in the synaptic cleft of the CNS and they are involved in several neurological diseases as well as cancer. Therefore, it is important to understand how these proteins work and how different ligands and inhibitors bind to these transporters. A few structures of the glutamate transporter family have been already published, and the complete transport mechanism of the SLC1A family is well known.

The authors describe 2 new Cryo-EM structures of EAAT2 in the presence of substrate and the inhibitor WAY-213613 and they reveal a new inhibitor binding site for this glutamate transporter family. The manuscript is well written and the methodology used is appropriate.

Although the authors present the structure of the human EAAT2 in the presence of glutamate and the inhibitor WAY-213613, no other experiment was performed to understand and validate the inhibition mechanism by WAY-213613. No mutagenesis analysis or functional analysis was done to corroborate the findings in the Cryo-EM structures. There is no validation in the paper, other than the structures, suggesting that this new inhibitor binding site is real.

This could be done, for example, by mutating key amino acids that interact with the inhibitor WAY-213613 to see if the inhibition could be abolished. This could be analysed either by binding experiments or structural determination. This would be a good way to validate the new inhibitor binding site.

I believe the validation reports are mixed up at my end. In the validation report that seems to correspond to the EAAT2-WAY213613, the structures of the ligands Y01 have issues in bond lengths and bond angles. As the paper's main outcome is a new inhibitor binding site of the EAATs, it would be good to see omit maps for the ligands to better appreciate the densities around them.

Another aspect that could be investigated is the difference between monomers. Have the authors observed the same density for the WAY-213613 in the three monomers forming the trimer? A symmetry

expansion and focused refinement could be performed to identify any structural variation between protomers.

Reviewer #1 (Remarks to the Author):

Zhang et. al. report human EAAT2 structures determined in glutamate and inhibitors bound states. These new structures are important, but the manuscript can be considerably improved with functional data to validate binding sites for glutamate and the inhibitor.

Reply: We appreciate reviewer's suggestions for the improvement of our manuscript. We have performed electrophysiological experiments with wild-type hEAAT2 and mutants transiently transfected HEK293T cells to validate the binding sites for the substrate glutamate and the inhibitor WAY-213613.

We supplemented functional characterization of hEAAT2 in revised manuscript. In the lines 75-87 it reads "In addition to mediating excitatory amino acid transport, hEAAT2 also acts as an anion-selective channel²⁷⁻²⁹. The hEAAT2-associated current includes three components: glutamate-induced anion current, Na⁺-dependent anion leak current and glutamate transport current^{30,31}. The competitive inhibitors can block all three components of the current³². To gain more insights into glutamate and inhibitor binding sites, we carried out electrophysiological experiments using whole cell patch-clamp technique under similar conditions to the ones described previously²⁵. In line with previous reports^{31,33,25}, application of substrate glutamate increased the amplitudes of anion leak currents (Supplementary Fig. 1a). In contrast, the WAY-213613 was able to block the anion leak conductance (Supplementary Fig. 1b). The activation of anion leak currents mediated by glutamate and inhibition of the ones by WAY-213613 were both dose dependent and could be fitted into a Michaelis-Menten-type equation, yielding a K_m of $30 \pm 2.5 \mu\text{M}$ for glutamate and an apparent K_i of $0.07 \pm 0.03 \mu\text{M}$ for WAY-213613 (Fig. 1a-1b), both of which are consistent with previous reports^{34,26}."

We supplemented a short discussion about the functional analysis for the residues D475 and R478 in revised manuscript. In the lines 142-153 it reads "To validate the glutamate binding site, we mutated D475^{TM8} and R478^{TM8} to alanine separately (D475A^{TM8} and R478A^{TM8}). Electrophysiological experiments showed that glutamate failed to activate anion currents of these two mutants at the concentration of glutamate up to 1 mM, as compared with that of wild-type hEAAT2 (Fig. 2e). To rule out the possibility that mutation in either residue might disrupt the activity of hEAAT2 as an anion channel, we also tested the inhibitory effects of WAY-213613 on anion currents of these two mutants. As a result, both mutants, D475A^{TM8} and R478A^{TM8}, still could mediate an inward anion current, evidenced by an apparent outward current induced by the WAY-213613, although the apparent K_i of WAY-213613 was reduced to $30.48 \mu\text{M}$ and $1.68 \mu\text{M}$ for D475A^{TM8} and R478A^{TM8}, respectively (Fig. 2f). Therefore, we speculate that the residues D475^{TM8} and R478^{TM8} are critical for the binding of hEAAT2 with glutamate and might also participate in the binding of this transporter with WAY-213613."

Similarly, we supplemented a short discussion about the functional analysis for the residue S441 in revised manuscript. In the lines 165-178, it reads "To investigate the functional role of the S441^{HP2}, we substituted this residue with glycine (S441G^{HP2}) and

performed electrophysiological recordings by applications of various concentrations of glutamate. Strikingly, we found that the S441G^{HP2} mutant displays higher sensitivity to glutamate with the K_m at $0.40 \pm 0.03 \mu\text{M}$ (Fig. 2h), which is increased by ~77-fold higher than that of wild-type hEAAT2, although the S441^{HP2} is not directly involved in the glutamate binding (Fig. 2c and 2d). This residue is exclusively present in the hEAAT2, as compared with the conserved glycine at the corresponding position in other homologs (Supplementary Fig. 7). We speculate that the unique residue S441^{HP2} probably provides additional interactions to stabilize the HP2 at a sealed conformation and prohibit glutamate release from the intracellular side. Thus, mutation in the S441^{HP2} might rupture these interactions, facilitate the glutamate release, accelerate the glutamate uptake cycle, and consequently biases the hEAAT2 towards a channel with higher open probability. The apparent K_i of WAY-213613 for the mutant S441G^{HP2} was reduced to $0.26 \mu\text{M}$ (Fig. 2f), which hints that this site also affects the binding of the inhibitor in a certain way.”

Figure 1*. New Fig. 2e, 2f and 2h in revised manuscript. **e.** Comparison of the glutamate-related current densities between the tested mutants and wild-type hEAAT2. Currents were recorded at $1000 \mu\text{M}$ glutamate and presented after Na^+ dependent leak current was subtracted. Sample sizes (n) tested for wild-type hEAAT2, D475A^{TM8} or R478A^{TM8} are 5, 4, 5 cells. **f.** WAY-213613 dose-response relationships for the D475A^{TM8}, R478A^{TM8} or S441G^{HP2} mediated currents. WAY-213613 was varied at the following concentrations: $0.3 \mu\text{M}$, $1 \mu\text{M}$, $3 \mu\text{M}$, $10 \mu\text{M}$, $30 \mu\text{M}$, $100 \mu\text{M}$ for D475A^{TM8} and R478A^{TM8}; $0.03 \mu\text{M}$, $0.1 \mu\text{M}$, $0.3 \mu\text{M}$, $1 \mu\text{M}$, $3 \mu\text{M}$, $10 \mu\text{M}$, $30 \mu\text{M}$ for S441G^{HP2}. Currents were normalized to the maximal current recorded after application of $30 \mu\text{M}$ or $100 \mu\text{M}$ WAY-213613. Sample sizes (n) tested from low to high concentrations are listed as follows: $n=4, 5, 5, 5, 9$ cells for D475A^{TM8}; $n=5, 5, 5, 4, 5, 12$ cells for R478A^{TM8}; $n=4, 4, 4, 5, 5, 4, 7$ cells for S441G^{HP2}. The lines represent the best fits to a Michaelis-Menten-like equation. The data of wild-type hEAAT2 from Fig. 1b was compared with those of the tested mutants. **h.** Glutamate dose-response relationship for the S441G^{HP2} mediated currents at the varied concentrations of $0.1 \mu\text{M}$, $0.3 \mu\text{M}$, $1 \mu\text{M}$, $3 \mu\text{M}$, $10 \mu\text{M}$, $30 \mu\text{M}$. The lines represent the best fit to a Michaelis-Menten-like equation with an average apparent K_m of $0.40 \pm 0.03 \mu\text{M}$. Currents were normalized to the maximal current recorded after application of $30 \mu\text{M}$ glutamate. Sample sizes (n) tested from low to high concentrations are 4, 5, 5, 5, 12, 5 cells.

Also, we supplemented a short discussion about the functional analysis for the residues I464, L467, and V468 in revised manuscript. In the lines 237-245, it reads “To validate the above hypotheses, we designed three mutations, including I464V^{TM8}, L467I^{TM8} and V468I^{TM8}. The electrophysiological experiments indicate that mutations do not affect glutamate binding. For these three mutants, application of glutamate in the external solution can significantly activate anion current with similar efficacy as that of wild-type

hEAAT2 (Fig. 4d). However, these mutations lead to a significant decrease in sensitivity to WAY-213613, with K_i increased to $\sim 0.17 \mu\text{M}$, $\sim 0.46 \mu\text{M}$ and $\sim 0.20 \mu\text{M}$ for I464V^{TM8}, L467I^{TM8} and V468I^{TM8} mutants, respectively (Fig. 4e), supporting our speculations that the I464^{TM8}, L467^{TM8} and V468^{TM8} residues are crucial for the binding specificity of hEAAT2 with WAY-213613.”.

Figure 2*. New Fig. 4d and 4e in revised manuscript. **d.** Glutamate dose-response relationships for the I464V^{TM8}, L467I^{TM8} or V468I^{TM8} mediated currents. The dose-response relationships for the tested mutants were analyzed at the varied concentrations of glutamate indicated in Fig. 1a. Sample sizes (n) tested from low to high concentrations are listed as follows: n=5, 5, 5, 5, 5, 5 cells for the mutant I464V^{TM8}; n=4, 4, 5, 5, 5, 5 cells for the mutant L467I^{TM8}; n=5, 5, 5, 5, 4, 6 cells for the mutant V468I^{TM8}. **e.** WAY-213613 dose-response relationships for the I464V^{TM8}, L467I^{TM8} or V468I^{TM8} mediated currents. The dose-response relationships for the hEAAT2 mutants as WAY-213613 was varied at the indicated concentrations in Fig. 1b. The lines represent the best fits to a Michaelis-Menten-like equation with apparent K_i of 0.166 μM , 0.464 μM , 0.196 μM , respectively. Currents were normalized to the maximal current recorded after application of 30 μM WAY-213613. Sample sizes (n) tested from low to high concentrations are: n=5, 5, 5, 4, 4, 5, 10 cells for I464V^{TM8}; n=5, 5, 5, 5, 5, 4, 11 cells for L467I^{TM8}; n=5,5,5,5,5,5,11 cells for V468I^{TM8}. The data of wild-type hEAAT2 from Fig. 1b was compared with those of the tested mutants.

Minor comments:

1. The manuscript appears to be cursorily written and can be improved. For example, in line 50, hASCT1 and hASCT2 do not transport glutamate.

Reply: Thank you for pointing this out. We revised the sentence in revised manuscript. In the lines 57-60, it now reads “The structures of archaeal homologs (Glt_{Ph}^{4,9,10,14,15}, Glt_{Tk}¹⁶⁻¹⁸) and four human SLC1 transporters (hEAAT1¹⁹, hEAAT3²⁰, hASCT1²¹, and hASCT2²²⁻²⁵) have been solved, and they share high sequence conservation in the ligand binding pocket including TM7, TM8, HP1 and HP2.”.

2. The authors may also want to compare EAAT2/glutamate with other substrate bound EAAT structures.

Reply: We thank reviewer’s suggestion. We compared the structures between the outward-facing hEAAT1^{Asp} and the inward-facing hEAAT2^{Glu} in the lines 123-129 of the main text. It now reads “Using the scaffold domain as a reference, we performed structural comparison between the outward-facing hEAAT1^{Asp} (PDB ID: 5LLU)¹⁹ and the

hEAAT2^{Glu} structures and found that the transport domain of hEAAT2^{Glu} structure is displaced by ~17 Å across the membrane towards cytoplasmic side relative to that of hEAAT1^{Asp}, suggesting that the hEAAT2^{Glu} structure is determined in an inward-facing conformation (Supplementary Fig. 5). Such conformational change of the transport domain is also consistent with previous observations²⁴.”

We supplemented a supplementary Fig. 5 for these comparisons and attached here for your convenience (Figure 3*).

Figure 3*. New Supplementary Fig. 5 in revised manuscript. **a.** Conformational change between the transport domain of the inward-facing hEAAT2^{Glu} (pale green) and that of the outward-facing hEAAT1^{Asp} (light blue), using the scaffold domain (wheat) as a reference. The HP1 and HP2 are colored in blue and red, respectively. The substrate of the hEAAT1^{Asp} (left panel) and that of the hEAAT2^{Glu} (right panel) are shown as colored spheres. **b.** Superimposed structures of the inward-facing hEAAT2^{Glu} (pale green) and the outward-facing hEAAT1^{Asp} (light blue).

We also have compared the structures between hEAAT2 and eukaryotic homologs (hEAAT1, hEAAT3, hASCT2) and prokaryotic homologs (Glt_{Ph}, Glt_{Tk}) in the lines 136-142 of the main text. It now reads “By comparison with eukaryotic homologs (hEAAT1¹⁹, hEAAT3²⁰, hASCT2²²) and prokaryotic homologs (Glt_{Ph}¹⁴, Glt_{Tk}¹⁷), key residues involved in substrate binding are highly conserved between hEAAT2 and the above-mentioned homologs (Supplementary Fig. 6). However, R478^{TM8} in the hEAAT2 is substituted by C467^{TM8} at the corresponding position in the neutral amino acid transporter hASCT2 (Supplementary Fig. 6e), which contributes to the substrate selectivity of hASCT2 for neutral amino acids²⁰.”

We supplemented a supplementary Fig. 6 for these comparisons and attached here for your convenience (Figure 4*).

Figure 4*. New Supplementary Fig. 6 in revised manuscript. **a.** Structural comparison of the transport domains between hEAAT2^{Glu} (pale green) and hEAAT1^{Asp} (PDB ID: 5LLU, light blue). **b.** Structural comparison of the transport domains between hEAAT2^{Glu} (pale green) and hEAAT3^{Asp} (PDB ID: 6X2Z, wheat). **c.** Structural comparison of the transport domains between hEAAT2^{Glu} (pale green) and Glt_{PH}^{Asp} (PDB ID: 6X15, white). **d.** Structural comparison of the transport domains between hEAAT2^{Glu} (pale green) and Glt_{TK}^{Asp} (PDB ID: 6R7R light pink). **e.** Structural comparison of the transport domains between hEAAT2^{Glu} (pale green) and hASCT2^{Asp} (PDB ID: 6GCT, salmon). In Supplementary Fig. 6a-e, the HP1 tip of hEAAT2^{Glu} is highlighted in blue to be differentiated from those of other homologs.

3. Since both EAAT/glutamate and EAAT/Asp structures are available, can these structures explain why EAAT prefers Glu over Asp (or lack of selectivity).

Reply: We appreciate reviewer's comment. Previous reports showed that the K_m values of hEAAT1, hEAAT2, and hEAAT3 for glutamate were 48 μ M, 97 μ M and 62 μ M, respectively; the K_m values of hEAAT1, hEAAT2 and hEAAT3 for aspartate were 60 μ M, 54 μ M and 47 μ M, respectively (PMID: 7521911). Therefore, the hEAATs do not exhibit a significant difference in substrate selectivity between glutamate and aspartate. Our structural comparison also supports these biochemical experimental results, since the key residues involved in substrate binding are highly conserved (Figure 4*).

4. In Figure 3d, Leu447 should be Leu467?

Reply: Thank you very much for the reminder. We have updated the Fig. 3d and added the residue Leu467 in the revised Fig. 3d. Moreover, we also adjusted the orientation of

the WAY-213613. The new orientation is similar to the one in the Fig. 3c.

Figure 5*. Old and revised version of Fig. 3d. **a.** Old version of Fig. 3d. **b.** Revised version of Fig. 3d.

Reviewer #2 (Remarks to the Author):

SUMMARY

In this manuscript, Zhang et al. solve the structures of human EAAT2 in complex with its substrate glutamate and inhibitor called WAY-213613 using cryo-EM. The good resolution of these structures allows for both the substrate and inhibitor to be placed in the map and consequently the mechanism of inhibition by WAY-213613 to be delineated as competitive antagonism.

Reply: We appreciate reviewer's positive comments and suggestions for improving our manuscript. We have made revisions as described below.

MAJOR COMMENTS

1. It is mentioned that inhibitors of EAAT2 may be an important therapeutic target, but the discussion needs more depth. It seems that the authors are claiming that inhibition of EAAT2 may be clinically however this reviewer questions the usefulness of EAAT2 inhibitors clinically given that deficiencies in EAAT2 cause disease. The authors point to Ref 11 for evidence that EAAT2 inhibitors may be clinically useful however this is actually not mentioned in the paper referred to at all.

Reply: Thank you for pointing out this confusing statement. We factually want to express that hEAAT2, but not inhibitors of hEAAT2, is an important therapeutic target. We have clarified this in the lines 45-48 of revised manuscript. Now it reads "Deficiency of hEAAT2 causes progressive neuronal death, and psychiatric or neurological diseases, including major depressive disorder, epilepsy, Alzheimer's disease, stroke, Parkinson's disease and amyotrophic lateral sclerosis (ALS), and thus hEAAT2 represents a potential

therapeutic target⁶⁻⁸.”.

2. While the structures provided by Zhang et al. provide significant insight as to the mechanism of inhibition by WAY-213613, for this paper to be complete, additional experiments are required that delineate a structure activity relationship. Functional experiments investigating the residues that coordinate WAY-213613 need to be conducted. For example, mutating these residues to the equivalent residues in other EAATs that have a lower affinity for WAY-213613 will help pinpoint exactly what residues are critical for the specific inhibition of EAAT2.

Reply: We appreciate reviewer’s constructive suggestion about the functional experiments. We have performed electrophysiological experiments for wild-type hEAAT2 and mutants to validate these key residues coordinating with WAY-213613. It turns out that the I464, L467, and V468 residues from TM8 play pivotal roles in the binding hEAAT2 with WAY-213613. We have added a brief discussion about the functional analysis in the lines 237-245 of revised manuscript, it reads “To validate the above hypotheses, we designed three mutations, including I464V^{TM8}, L467I^{TM8} and V468I^{TM8}. The electrophysiological experiments indicate that mutations do not affect glutamate binding. For these three mutants, application of glutamate in the external solution can significantly activate anion current with similar efficacy as that of wild-type hEAAT2 (Fig. 4d). However, these mutations lead to a significant decrease in sensitivity to WAY-213613, with K_i increased to ~0.17 μ M, ~0.46 μ M and ~0.20 μ M for I464V^{TM8}, L467I^{TM8} and V468I^{TM8} mutants, respectively (Fig. 4e), supporting our speculations that the I464^{TM8}, L464^{TM8} and V468^{TM8} residues are crucial for the binding specificity of hEAAT2 with WAY-213613.”.

We also attached related functional data in this response file for your convenience just as shown in Figure 2*.

3. The cryoEM maps provided show clear density for glutamate and WAY-213613. Interestingly, they also show density for a DDM molecule in the glutamate binding site in the WAY-213613-bound structure. This reviewer feels that this is a very interesting observation that warrants discussion in the manuscript. Could it be that inhibition by WAY-213613 causes other molecules such as detergents – or more physiologically relevant – lipids to interact with residues in the glutamate binding site and also contributing to inhibition of transport?

Reply: We appreciate reviewer’s comment and agree with the reviewer that the lipid molecule may contribute to the inhibition of glutamate transport by WAY-213613 across the membrane. We also briefly discussed this structural observation in the lines 200-207 of revised manuscript as reviewer suggested. It reads “In the cryo-EM map of hEAAT2^W complex, a DDM-like molecule was found to be sandwiched between TM1 and the unwound helix of TM8. Its hydrophobic tail points towards the extracellular side. The head group is positioned proximal to the bromofluorophenol group and form hydrophobic interactions to stabilize the WAY-213613 binding (Supplementary Fig. 4b and 4e). Therefore, we speculate that a lipid molecule in the membrane environment may occupy the binding position of the DDM and participate in the inhibitory effects of glutamate

transport by the WAY-213613.”.

Figure 6*. New Supplementary Fig. 4e in revised manuscript. **e.** Density of DDM in the hEAAT2^W complex. DDM is colored in cyan and the density of DDM is shown as mesh.

4. This manuscript lacks any discussion of the results or a conclusion – these are essential aspects to scientific literature and without them this manuscript is incomplete. This reviewer feels especially strongly about this given that inhibition of glutamate transporters therapeutically is somewhat controversial and warrants, at the very least, further thought and discussion.

Reply: We appreciate reviewer’s comments and we have supplemented a discussion in the lines 235-301 of revised manuscript. It reads “hEAAT2 is the most abundantly expressed excitatory amino acid transporter in astrocytes, which is obligate for the uptake of glutamate from the synaptic cleft into astrocytes to prevent neuronal cytotoxicity³. Here, we report the cryo-EM structure of hEAAT2^{Glu} in complex with glutamate, which reveals the molecular details of how hEAAT2 recognizes the glutamate. By comparison with eukaryotic homologs (hEAAT1¹⁹, hEAAT3²⁰, hASCT2²²) and prokaryotic homologs (Glt_{Ph}¹⁴, Glt_{Tk}¹⁷) (Supplementary Fig. 6), the key interaction residues involved in substrate binding are highly conserved between hEAAT2 and the above-mentioned homologs. Meanwhile, we found that a unique residue S441^{HP2} is exclusively present in hEAAT2 and it is substituted by the conserved glycine at the corresponding position of other homologs (Supplementary Fig. 7). Electrophysiological experiments showed that the affinity of S441G^{HP2} to glutamate can be significantly increased as compared with that of wild-type hEAAT2 (Fig. 2h). Notably, the S441^{HP2} is located at the HP2 tip, and does not directly take part in the interaction with glutamate (Fig. 2g). Thus, we speculate that S441^{HP2} may affect the rate of transport glutamate. Further experiments will be required to clarify the functional roles of S441^{HP2} in hEAAT2.

In this study, we also resolved the structure of hEAAT2 in complex with the inhibitor WAY-213613, which clearly elucidates the binding pocket of hEAAT2 for WAY-213613. The hEAAT2^W complex is stabilized at the inward-facing conformational state. The α -carboxyl and α -amino and the aniline groups of WAY-213613 form contacts with residues S364^{HP1},

D475^{TM8} and R478^{TM8} and share an overlapped binding site with the substrate glutamate, in line with a notion that WAY-213613 is a competitive inhibitor. In the hEAAT2^W complex, the HP2 undergoes a remarkably conformational change and rotates away from the HP1, thus creates enough space for the WAY-213613 binding. Interestingly, the S441^{HP2} was determined to form a hydrogen bond with T395^{TM7} and consequently stabilize the HP2 in an open conformation, which turns out that this interaction is critical for the WAY-213613 binding (Fig. 3c and 2f). Meanwhile, the bromofluorophenol group of WAY-213613 is wrapped by some hydrophobic residues from TM8, TM7b and HP2b, such as I464^{TM8}, L467^{TM8}, V468^{TM8}, M450^{HP2} and L447^{HP2} (Fig. 3c). Based on structural comparison and sequence alignment, the residues I464^{TM8}, L467^{TM8} and V468^{TM8} from TM8 in hEAAT2 are varied as compared with the corresponding residues of other hEAATs (Fig. 4c). The electrophysiological experiments demonstrate that the above three residues are critical for the inhibition of hEAAT2 by WAY-213613 with high potency and mutation in each of three residues substantially reduces the inhibitory efficiency of hEAAT2 by WAY-213613 (Fig. 4e). Considering the I464-L467-V468 cluster in hEAAT2 is substituted by Val-Leu-Ile in hEAAT3 or Ile-Ile-Val in hEAAT1/hEAAT4/hEAAT5, respectively, we speculate that the I464-L467-V468 cluster from TM8 acts as a key structural determinant for the selective inhibition of hEAAT2 by WAY-213613.”.

MINOR COMMENTS

- Line 25: “acts” should be “serves”.

Reply: We thank reviewer’s suggestion and have made the correction as “Glutamate is the predominant excitatory neurotransmitter, which serves as a key role in the development of the mammalian central nervous system, and participates in normal brain function, such as incorporating learning, cognition and memory¹.” in the lines 32-34 of revised manuscript.

- Line 26: Remove second instance of “the”.

Reply: We appreciate this comment and “the” in the second instance has been removed from the corresponding position in revised manuscript.

- Line 27: It should be noted that excessive concentrations are neurotoxic.

Reply: We thank reviewer’s suggestion and corrected the corresponding description as “Also, excessive glutamate can lead to excitotoxicity, which may kill neuronal cells through excessive stimulation of glutamate receptors².” in the lines 34-36 of revised manuscript.

- Line 32: “Buffering glutamate” doesn’t really make sense. Should be replaced with “transporting glutamates back up into the pre-synapse or astrocytes”.

Reply: We thank reviewer’s suggestion and have corrected the sentence in the lines 37-41 of revised manuscript. Now it reads “EAATs are known as excitatory amino acid transporters including five subtypes (EAAT1–EAAT5), which are responsible for the removal of glutamate from the synaptic cleft by rapidly binding and transporting

glutamates back up into the pre-synapse or astrocytes, which contributes to the termination of synaptic activity and to the clearance of potentially cytotoxic extracellular glutamate³.”.

- Lines 39-40: As noted above in major comments, more detail is warranted to explain how EAAT2 may be an important therapeutic target.

Reply: We thank reviewer’s comment. We have added more details in the lines 45-48 of revised manuscript to justify hEAAT2 may be an important therapeutic target. It reads “Deficiency of hEAAT2 causes progressive neuronal death, and psychiatric or neurological diseases, including major depressive disorder, epilepsy, Alzheimer’s disease, stroke, Parkinson’s disease and amyotrophic lateral sclerosis (ALS), and thus hEAAT2 represents a potential therapeutic target⁶⁻⁸.”.

- Line 41: Ph for GltPh should be in subscript.

Reply: We appreciate very much for reviewer’s suggestions. We have checked throughout the manuscript and all “GltPh” has been corrected as “Glt_{Ph}” in revised manuscript. Moreover, we have also changed all “GltTk” to “Glt_{Tk}”.

- Line 47: The substrate binding site is described as being “close to HP1 and HP2”. There are several substrate-bound structures of glutamate transporters and given that this is a key finding of this paper, what is already known about the substrate binding site should not be glazed over. More detail is required.

Reply: We thank reviewer’s suggestions. We have included more details about substrate binding site in revised manuscript (lines 53-56). It reads “Previous studies have shown that the individual subunit in EAATs can transport the substrate independently¹¹⁻¹³, and the substrate binding site is constituted by the central unwound region of TM7 (NMDGT motif), TM8 and tips of HP1 and HP2^{3,9}.”.

- Line 52: What is the percentage identity between human EAAT2 and other human EAATs? How much does it vary in functionally important regions?

Reply: We appreciate reviewer’s comments. We carried out sequence alignment between hEAAT2 and its homologs and hEAAT2 shares 34% sequence identity with Glt_{Ph} and Glt_{Tk}, 50% sequence identity with hEAAT1 and hEAAT3, 42% and 39% sequence identity with hASCT1 and hASCT2, respectively. The substrate binding pocket including TM7, TM8, HP1 and HP2 shares high sequence conservation between hEAAT2 and its homologs (Supplementary Fig. 7). We added sequence identity information in revised manuscript (line 60-63). It reads “However, it is still desirable to determine the structure of hEAAT2, as it shares low sequence identities with these homologs (34% sequence identity with Glt_{Ph} and Glt_{Tk}, 50% sequence identity with hEAAT1 and hEAAT3, 42% and 39% sequence identity with hASCT1 and hASCT2, respectively).”.

- Line 53: It would be useful at this point of the paper to know the affinities of WAY-213613 for different human EAATs rather than at line 157.

Reply: We thank reviewer’s suggestion. We have introduced the affinity information of

WAY-213613 in the lines 63-65 of the revised introduction. It reads “WAY-213613 is a potent and highly selective inhibitor for hEAAT2 (IC₅₀ is 85 nM), which has 59-fold and 44-fold affinity over those of hEAAT1 and hEAAT3 (IC₅₀ is 5 and 3.8 μM, respectively)²⁶.”.

• Supplementary Figures 2 and 3: It is difficult to see the densities. Please change the colors. Also, the densities for substrates and inhibitors should be shown in these figures.

Reply: We thank reviewer’s comment. We revised the figures for clarification. The revised Supplementary Fig. 2e and S3e are attached here for your convenience (Figure 7*).

We have also included the ligand density in the main figures as suggested by another reviewer. We attached the revised figures in this response file (see Figure 10* and Figure 12*).

Figure 7*. Revised version of **Supplemental Fig. 2 and 3**. **a.** Revised version of Supplementary Fig. 2e. **b.** Revised version of Supplementary Fig. 3e.

• Lines 67-68: References are missing for the structures referred to.

Reply: We thank reviewer’s comment. We have added proper citations in the lines 91-93

in the revision as “hEAAT2 has a homotrimer structure resembling other EAATs (hEAAT1¹⁹ and hEAAT3²⁰) and its orthologs (hASCT1²¹ and hASCT2^{22–25}) or paralogs (Glt_{Ph}^{4,9,10,14,15} and Glt_{Tk}^{16,18}).”.

• Line 76: What do the authors mean by “HP1 is attached to the inner membrane”? This reviewer is not familiar with the concept of a protein attaching to the membrane.

Reply: We thank reviewer’s comment. We have corrected the corresponding description as “In our hEAAT2 structures, the HP1 is situated approximately parallel to the membrane plane and almost all exposed in the cytoplasm (Fig. 1d).” in the lines 101-102 of revised manuscript.

• Supplementary Figure 4 and Lines 86-88: While it has been described that CHS-resembling densities could be identified, this is not clearly demonstrated in Supplementary Figure 4. The authors should show how CHS fits into the density with more detail if this is their claim.

Reply: We thank reviewer’s comment. We have corrected the depiction of CHS in the lines 112-117 of revised manuscript as “In fact, CHS resembling densities are determined to be located in the cavity formed by TM3, TM6 and TM8 at the inner lobe of plasma membrane (Supplementary Fig. 4c and 4d), which was also found in the maps of hEAAT3³⁵ and Glt_{Ph}¹⁴ (6S3Q and 6X15, respectively), suggesting that cholesterol may be correlated to the assembly or function of SLC1 transporters and their homologs.”.

Figure 8*. New Supplementary Fig. 4c and S4d in revised manuscript. c. and d. Density of CHS in hEAAT2^{Glu} and hEAAT2^W complex, respectively. CHS is colored in yellow and the densities are shown as meshes.

• Supplementary Figure 4: The colors for the hairpins are only shown in one of three protomers. Either say this in the legend or show it in all three protomers. Also, HP2 is listed as being blue when really it looks purple. Furthermore, WAY-213613 is said to be colored in pink, when it is actually red, and this can be very confusing with the pink-magenta used to color HP2 in this panel. Please use more distinct and consistent colors

for these figures. Labels may also be helpful.

Reply: We thank reviewer's comment and feel sorry for the improper color scheme. We updated Supplementary Fig. 4 and the corresponding figure legends for clarity in revised manuscript.

We also attached the revised Supplementary Fig. 4 here (Figure 9*).

Figure 9*. Revised version of Supplemental Fig. 4. **a.** and **b.** Cryo-EM maps of the hEAAT2^{Glu} and hEAAT2^W complexes. The homotrimer is viewed from the cytoplasm (left panel) and membrane plane (right panel), respectively. The scaffold domain and the transport domain are colored in wheat and green, respectively. HP1 and HP2 are colored in blue and red, respectively. CHS is highlighted in a dashed-line rectangular box. The densities of WAY-213613 and DDM are colored in pink and cyan, respectively. Lipid densities observed around the complex are highlighted in orange.

• Lines 93-95: This sentence seems out of place and is unnecessary at this point in the manuscript.

Reply: We thank reviewer's comment. We have removed this sentence from revised manuscript.

• Figure 2: I see beta sheets in the structure, but no beta sheets have been mentioned. Please explain.

Reply: We thank reviewer's comment. We have added these beta sheets information in the lines 95-97 of revised manuscript. Now it reads "These secondary structures assemble into a scaffold domain (TM1, 2, 4, 5 and two extracellular beta sheets insert in TM4b and TM4c) and a transport domain (TM3, 6-8 and HP1, HP2), respectively."

• **Figure 2:** Density for glutamate should be shown in a close-up view.

Reply: We thank reviewer's comment. We updated Fig. 2 with added a close-up view for glutamate into the panel b for clarity in revised manuscript.

Figure 10*. Old and revised version of Fig. 2b. **a.** Old version of Fig. 2b. **b.** Revised version of Fig. 2b.

• **Lines 101-107:** How does this substrate binding site compare to other substrate-bound structures? This point warrants discussion and prior work should be acknowledged here.

Reply: We thank reviewer's suggestion. We have added the descriptions of the structural comparison between hEAAT2 and eukaryotic homologs (hEAAT1, hEAAT3, hASCT2) and prokaryotic homologs (Glt_{Ph}, Glt_{Tk}) in the main text.

In the lines 136-142, it now reads "By comparison with eukaryotic homologs (hEAAT1¹⁹, hEAAT3²⁰, hASCT2²²) and prokaryotic homologs (Glt_{Ph}¹⁴, Glt_{Tk}¹⁷), key residues involved in substrate binding are highly conserved between hEAAT2 and the above-mentioned homologs (Supplementary Fig. 6). However, R478^{TM8} in the hEAAT2 is substituted by C467^{TM8} at the corresponding position in the neutral amino acid transporter hASCT2 (Supplementary Fig. 6e), which contributes to the substrate selectivity of hASCT2 for neutral amino acids²⁰."

We have also presented a new Supplementary Fig. 6 and we attached it as Figure 4* in this response.

• **Line 109-110:** "Upon the glutamate binding" should be replaced with "when glutamate is bound" because here, we are talking about the release of substrate.

Reply: We thank reviewer's comment and have corrected this sentence as reviewer suggested (lines 154-156). It now reads "Compared with inward-facing hEAAT3 at apo

state (PDB ID: 6X3F)³⁶, we found that the HP2 in the hEAAT2^{Glu} complex remarkably shifts towards the intracellular side when glutamate is bound (Fig. 2g), which would prevent the escape of substrate.”

• Lines 115-118: Is S441 important for substrate transport? This warrants functional investigation, especially since it is not conserved amongst the different homologues. Why are these interactions of S441 significant?

Reply: We appreciate reviewer’s comment. We have performed electrophysiological experiments to explore the functional roles of the S441 and added a discussion in the lines 165-178 of revised manuscript. It reads “To investigate the functional role of the S441^{HP2}, we substituted this residue with glycine (S441G^{HP2}) and performed electrophysiological recordings by applications of various concentrations of glutamate. Strikingly, we found that the S441G^{HP2} mutant displays higher sensitivity to glutamate with the K_m at $0.40 \pm 0.03 \mu\text{M}$ (Fig. 2h), which is increased by ~77-fold higher than that of wild-type hEAAT2, although the S441^{HP2} is not directly involved in the glutamate binding (Fig. 2c and 2d). This residue is exclusively present in the hEAAT2, as compared with the conserved glycine at the corresponding position in other homologs (Supplementary Fig. 7). We speculate that the unique residue S441^{HP2} probably provides additional interactions to stabilize the HP2 at a sealed conformation and prohibit glutamate release from the intracellular side. Thus, mutation in the S441^{HP2} might rupture these interactions, facilitate the glutamate release, accelerate the glutamate uptake cycle, and consequently biases the hEAAT2 towards a channel with higher open probability. The apparent K_i of WAY-213613 for the mutant S441G^{HP2} was reduced to $0.26 \mu\text{M}$ (Fig. 2f), which hints that this site also affects the binding of the inhibitor in a certain way.”

We attached the related functional data here for your convenience (Figure 11*).

Figure 11*. New Fig. 2f and 2h in revised manuscript. f. WAY-213613 dose-response relationships for the D475A^{TM8}, R478A^{TM8} or S441G^{HP2} mediated currents. WAY-213613 was varied at the following concentrations: 0.3 μM , 1 μM , 3 μM , 10 μM , 30 μM , 100 μM for D475A^{TM8} and R478A^{TM8}; 0.03 μM , 0.1 μM , 0.3 μM , 1 μM , 3 μM , 10 μM , 30 μM for S441G^{HP2}. Currents were normalized to the maximal current recorded after application of 30 μM or 100 μM WAY-213613. Sample sizes (n) tested from low to high concentrations are listed as follows: n=4, 5, 5, 5, 9 cells for D475A^{TM8}; n=5, 5, 5, 4, 5, 12 cells for R478A^{TM8}; n=4, 4, 4, 5, 5, 4, 7 cells for S441G^{HP2}. The lines represent the best fits to a Michaelis-Menten-like equation. The data of wild-type hEAAT2 from Fig. 1b was compared with those of the tested

mutants. **h.** Glutamate dose-response relationship for the S441G^{HP2} mediated currents at the varied concentrations of 0.1 μM , 0.3 μM , 1 μM , 3 μM , 10 μM , 30 μM . The lines represent the best fit to a Michaelis-Menten-like equation with an average apparent K_m of $0.40 \pm 0.03 \mu\text{M}$. Currents were normalized to the maximal current recorded after application of 30 μM glutamate. Sample sizes (n) tested from low to high concentrations are 4, 5, 5, 5, 12, 5 cells.

• Lines 124-125: "WAY-213613 has no effect on ionotropic and metabotropic glutamate receptors and thus it's a potential tool for elucidating the function of hEAAT2." We already know the function of EAAT2; this is a vague sentence and does not tell the reader why they should care about understanding how WAY-213613 binds to EAAT2. Careful thought, consideration and explanation is warranted here.

Reply: We thank reviewer's comment. We agree with the reviewer that this sentence does not clearly show why we study structure of WAY-213613 bound hEAAT2. The WAY-213613 is a potent and highly selective inhibitor for hEAAT2 (IC_{50} is 85 nM), which has 59-fold and 44-fold affinity over those of hEAAT1 and hEAAT3 (IC_{50} is 5 and 3.8 μM , respectively). It is desirable to understand how the WAY-213613 specifically bind with hEAAT2.

We include the selectivity information in the lines 63-65 in the revised introduction as the reviewer 1 suggested. It reads "WAY-213613 is a potent and highly selective inhibitor for hEAAT2 (IC_{50} is 85 nM), which has 59-fold and 44-fold affinity over those of hEAAT1 and hEAAT3 (IC_{50} is 5 and 3.8 μM , respectively)²⁶."

In the lines 180-181, we modified the previous statement. It now reads "The WAY-213613 displays high selectivity for the hEAAT2, thus it is desirable to investigate its specificity differences between hEAATs."

• Line 128 and Figure 3B: A more zoomed-in figure of WAY-213613 in the density map should be shown.

Reply: We thank reviewer's comment. We updated Fig. 3 with added a close-up view for WAY-213613 into the panel b in revised manuscript. I attached the revised Fig. 3b here for your convenience.

Figure 12*. Old and revised version of Fig. 3b. **a.** Old version of Fig. 3b. **b.** Revised version of Fig. 3b.

- Figures 2D and 3D are mirror images of each other. Please amend for ease of interpretation.

Reply: We thank reviewer's comment. We updated Fig. 2d and Fig. 3d for ease of interpretation in revised manuscript. We attached the original and revised Fig. 2d and Fig. 3d here for your convenience.

Figure 13*. Old and revised version of Fig. 2d and Fig. 3d. **a.** Old version of Fig. 2d. **b.** Revised version of Fig. 2d. **c.** Old version of Fig. 3d. **d.** Revised version of Fig. 3d.

- In several figures, two structures are overlaid with one structure shown in gray and the other shown in green as detailed by the figure legend. In these figures, HP2 is colored in red, but it is not detailed in the figure legend which structure this red (HP2) belongs to. Please clarify.

Reply: We thank reviewer's comment. We updated the related Figure legends to clearly

identify the attribution of HP2 in revised manuscript.

- Supplementary Figure 6: Please put PDB IDs into figure legends.

Reply: We thank reviewer's reminder. We have added PDB IDs into the corresponding Figure legends.

- Lines 168-181: There are also structures of TBOA bound to GltPh in outward-facing structures. These should also be discussed; how do they fit into this story?

Reply: We appreciate reviewer's comment for improving our manuscript, we have included the outward-facing TBOA bound Glt_{Ph} structure in the lines 246-251 of revised manuscript. Now it reads "So far, a serial of structures of hEAAT2 homologs have been determined in the presence of the different kinds of inhibitors, including competitive inhibitors TBOA^{10,14,18}, TFB-TBOA^{14,19,35}, trans and cis isomers of *p*-OMe-azo-TBOA³⁷, Lc-BPE²⁵ and allosteric inhibitors UCPH-101¹⁹. Complex structures with inward-facing conformation were adopted for further comparison with hEAAT2^W (Supplementary Fig. 8a), comprising Glt_{Ph}^{TFB-TBOA} (PDB ID: 6X14)¹⁴, Glt_{Ph}^{TBOA} (PDB ID: 6X16)¹⁴ and hEAAT3^{TFB-TBOA} (PDB ID: 6S3Q)³⁵."

Reviewer #3 (Remarks to the Author):

EAATS play a critical role in maintaining low glutamate concentrations in the synaptic cleft of the CNS and they are involved in several neurological diseases as well as cancer. Therefore, it is important to understand how these proteins work and how different ligands and inhibitors bind to these transporters. A few structures of the glutamate transporter family have been already published, and the complete transport mechanism of the SLC1A family is well known.

The authors describe 2 new Cryo-EM structures of EAAT2 in the presence of substrate and the inhibitor WAY-213613 and they reveal a new inhibitor binding site for this glutamate transporter family. The manuscript is well written and the methodology used is appropriate.

Although the authors present the structure of the human EAAT2 in the presence of glutamate and the inhibitor WAY-213613, no other experiment was performed to understand and validate the inhibition mechanism by WAY-213613. No mutagenesis analysis or functional analysis was done to corroborate the findings in the Cryo-EM structures. There is no validation in the paper, other than the structures, suggesting that this new inhibitor binding site is real.

This could be done, for example, by mutating key amino acids that interact with the inhibitor WAY-213613 to see if the inhibition could be abolished. This could be analysed either by binding experiments or structural determination. This would be a good way to validate the new inhibitor binding site.

Reply: We appreciate the reviewer's suggestion about functional experiments. We have

performed electrophysiological experiments with the mutants transiently transfected HEK293T to validate residues that coordinate with WAY-213613.

And we have supplemented the related content in revised manuscript. In the lines 237-245, it reads “To validate the above hypotheses, we designed three mutations, including I464V^{TM8}, L467I^{TM8} and V468I^{TM8}. The electrophysiological experiments indicate that mutations do not affect glutamate binding. For these three mutants, application of glutamate in the external solution can significantly activate anion current with similar efficacy as that of wild-type hEAAT2 (Fig. 4d). However, these mutations lead to a significant decrease in sensitivity to WAY-213613, with K_i increased to ~0.17 μ M, ~0.46 μ M and ~0.20 μ M for I464V^{TM8}, L467I^{TM8} and V468I^{TM8} mutants, respectively (Fig. 4e), supporting our speculations that the I464^{TM8}, L464^{TM8} and V468^{TM8} residues are crucial for the binding specificity of hEAAT2 with WAY-213613.”.

We also attached the related functional data in this response file (see Figure 2*).

I believe the validation reports are mixed up at my end. In the validation report that seems to correspond to the EAAT2-WAY213613, the structures of the ligands Y01 have issues in bond lengths and bond angles. As the paper's main outcome is a new inhibitor binding site of the EAATs, it would be good to see omit maps for the ligands to better appreciate the densities around them.

Reply: We thank the reviewer for pointing out this issue. We have refined the model. The bond lengths and bond angles of the Y01 molecule look good now and no outliers reported in the new validation reports. By the way, the Y01 is the CHS molecule, instead of the inhibitor WAY-213613.

For these cryo-EM structures, we could not generate omit maps for the ligand similarly to the routinely prepared ones using structures determined via X-ray crystallography. Moreover, both of our structures are bound with ligand glutamate or WAY-213613 and their binding site are overlapped, and thus we cannot prepare a difference map for either.

Another aspect that could be investigated is the difference between monomers. Have the authors observed the same density for the WAY-213613 in the three monomers forming the trimer? A symmetry expansion and focused refinement could be performed to identify any structural variation between protomers.

Reply: We appreciate reviewer's comment. The WAY-213613 was present during the whole course of sample preparation, including protein expression at the final concentration of 5 μ M for 60 h, protein purification at the final concentration of 5 μ M and also grids preparation at the final concentration of 200 μ M. Considering the high binding affinity of WAY-213613 to hEAAT2 (K_i ~80 nM), we believe that WAY-213613 is most likely to bind with all of three subunits. In our hEAAT2^W map, the WAY-213613 density in three subunits is nearly identical (Figure 14*), further supporting that hEAAT2 is fully occupied by the WAY-213613.

Figure 14*. Cryo-EM map of the WAY-213613 from three subunits. The WAY-213613 is shown as sticks, overlaid with corresponding density in blue meshes. The thresholds of these densities are same.

We also reprocessed our data using symmetry expansion and focused refinement as the reviewer suggested. After 3D classification, we got two major classes with secondary structure resolved. The Class-1 accounts for ~50% of total particles and further refinement yields a 3.6 Å map (Figure 15*a). This Class-1 map clearly shows the WAY-213613 density and that the overall structure is nearly identical with the original structure determined using C3 symmetry (hEAAT2-C3) (Figure 15*b). The second class (Class-2) occupies ~30% of total particles. The focused refinement gives rise to a 4.2 Å map (Figure 15*c). The map quality is worse than that of Class-1 and the drug density is weaker than the one in the Class-1 map (Figure 15*d). One possibility is that the Class-2 is stabilized at the apo state. Compared with the inward-facing hEAAT3 at the apo state (hEAAT3^{Apo}, PDB ID: 6X3F), we found that the HP2 in the Class-2 remarkably shifts away from the HP1 (Figure 15*e). The HP2 in the hEAAT3^{Apo} structure would generate steric clashes with WAY-213613 (hEAAT2^W complex) (Figure 15*f). However, the Class-2 adopts a similar conformation to that of the WAY-213613 bound Class-1 structure (Class-1^W), shows the HP2 tips at a same open conformation (Figure 15*g), suggesting that the Class-2 structure also harbors a binding pocket for WAY-213613. Considering a weak ligand density determined in the Class-2 map and the identical HP2 conformation with the one in the Class-1^W structure (Figure 15*d and 15*g), we speculate that the Class-2 structure may represent a drug bound state and the weak drug density in the Class-2 may be probably attributed to the map at a poor resolution caused by a small particle set for re-construction (~31 k particles).

Figure 15*. Structural analysis of hEAAT2 structures achieved using symmetry expansion. a. Cryo-EM map of Class-1^W with C1 symmetry imposed. **b.** Density of WAY-213613 in Class-1^W. **c.** Cryo-EM map of Class-2 with C1 symmetry imposed. **d.** Density of ligand in Class-2. In **Figure 15*a** and **c**, the homotrimer is viewed from the membrane plane (left panel) and cytoplasm (right panel), respectively. The scaffold domain and the transport domain are colored in wheat and green, respectively. HP1 and HP2 are colored in blue and red, respectively. The densities of WAY-213613 and DDM are colored in pink and cyan, respectively. Lipid densities observed around the complex are highlighted in orange. **e.** Structural comparison of the transport domains between the Class-2 and hEAAT3^{Apo} (6X3F). **f.** Structural comparison of the transport domains between the hEAAT2^W complex and hEAAT3^{Apo} (6X3F). **g.** Structural comparison of the transport domains between the Class-2 and Class-1^W.

REVIEWERS' COMMENTS

Reviewer #1 (Remarks to the Author):

The authors have improved the manuscript. I have some comments about the electrophysiology.

1. Could the authors provide current traces for Figure 2 and Figure 4 as supplementary data?
2. Could the authors provide more details about electrophysiology? For example, holding potential.
3. Could the authors explain why the currents remains throughout recording in Supplementary Figure 1? Were glutamate or inhibitors washed away toward the end of experiment, so currents returns to the baseline?

Reviewer #2 (Remarks to the Author):

This reviewer thanks the authors for the additional experiments, figures, sections of text, and other corrections, and is now satisfied with this manuscript for publication.

Reviewer #3 (Remarks to the Author):

The authors have adequately addressed all my questions.

The manuscript has considerably improved.

Reviewer #1 (Remarks to the Author):

The authors have improved the manuscript. I have some comments about the electrophysiology.

1. Could the authors provide current traces for Figure 2 and Figure 4 as supplementary data?

Reply: We thank reviewer's suggestion. We have provided the current traces for Figure 2 and Figure 4 as Supplementary Fig. 7 and 9 in our latest revised manuscript.

We attached the revised Supplementary Fig. 7 and 9 here for your convenience (Figure 1* and 2*, respectively).

Figure 1*. New Supplementary Fig. 7 in revised manuscript. **a-c.** Currents recorded when 1000 μM glutamate are applied to the hEAAT2 mutants (D475A^{TM8} and R478A^{TM8}) or wild-type hEAAT2-expressed HEK293T cells. **d.** Current traces obtained when glutamate was applied at the indicated concentrations to the hEAAT2 mutant S441G^{HP2}-expressed HEK293T cells. **e-g.** Current traces obtained when WAY-213613 was applied at the indicated concentrations to hEAAT2 mutants (D475A^{TM8}, R478A^{TM8} and S441G^{HP2})-expressed HEK293T cells. All experiments were executed at 0 mV and the experimental procedure was the same as the one described in the method.

Figure 2*. New Supplementary Fig. 9 in revised manuscript. **a-c.** Current traces obtained when glutamate was applied at the indicated concentrations to the hEAAT2 mutants (I464V^{TM8}, L467I^{TM8} and V468I^{TM8})-expressed HEK293T cells. **d-f.** Current traces obtained when WAY-213613 was applied at the indicated concentrations to hEAAT2 mutants (I464V^{TM8}, L467I^{TM8} and V468I^{TM8})-expressed HEK293T cells. All experiments were executed at 0 mV and the experimental procedure was the same as the one described in the method.

2. Could the authors provide more details about electrophysiology? For example, holding potential.

Reply: We thank the reviewer for this suggestion. We added more details into the updated method of electrophysiology. In the lines 430-434, it reads “All experiments were

executed at 0 mV. After the cell current was stabilized, glutamate or WAY-213613 was sustained at the indicated concentrations from external dosing for at least 6 seconds. Subsequently, the cell was washed away with the external buffer to start a new round of applications or end the experiment. The plateau currents were used for the following calculations.”.

3. Could the authors explain why the currents remains throughout recording in Supplementary Figure 1? Were glutamate or inhibitors washed away toward the end of experiment, so currents return to the baseline?

Reply: We thank the reviewer for this comment. The current recorded in the electrophysiological experiment is composed glutamate-induced anion current, which is elicited during the glutamate uptake cycle. The transporter at inward- or outward-facing conformation could not mediate anion current, and the anion conduction pathway can be formed in intermediate conformations (PMID: 25635461). Our electrophysiological experiments are performed with external buffer contained 140 mM NaCl, 2 mM MgCl₂, 2 mM CaCl₂, and 10 mM HEPES, and internal pipette solution comprised of 130 mM NaSCN, 2 mM MgCl₂, 10 mM EGTA, 10 mM glutamate, where hEAAT2 is in an exchange mode. When application of extracellular glutamate or WAY-213613 in the exchange mode can result in a redistribution of glutamate binding sites within the membrane associated with permanent activation of an anion current (PMID: 16478724, PMID: 11382805, PMID: 12451116). Thus, the currents can be remained throughout the recording in Supplementary Figure 1.

The current is able to return to the baseline, after we wash the ligand away. However, we did not record the current during the washing process. As shown in the figure3*, we used the same cell to test the glutamate or WAY-213613 dependent activity. Application of glutamate or WAY-213613 generates apparent inward or outward current, respectively. After the ligands was washed away with external buffer, we execute the next round of experiment with higher concentration of glutamate or WAY-213613. The onset of each trace recorded with different concentration of glutamate or WAY-213613 is same with each other, indicating the current returned to the baseline after removing the ligands.

Figure 3*. Functional characterization of wild-type hEAAT2 samples. a. Current traces obtained when 3 μM , 10 μM , 30 μM , 100 μM , 300 μM , 1000 μM glutamate are applied to the hEAAT2-expressed HEK293T cells. **b.** Current traces obtained when 0.1 μM , 0.3 μM , 1 μM , 3 μM , 30 μM WAY-213613 are applied to hEAAT2-expressed HEK293T cells.

Reviewer #2 (Remarks to the Author):

This reviewer thanks the authors for the additional experiments, figures, sections of text, and other corrections, and is now satisfied with this manuscript for publication.

Reviewer #3 (Remarks to the Author):

The authors have adequately addressed all my questions.
The manuscript has considerably improved.